# Anomalous enhancement of thermoelectric power factor in multiple two-dimensional electron gas system

Yuto Uematsu [1], Takafumi Ishibe [1], Takaaki Mano [2], Akihiro Ohtake [2], Hideki T. Miyazaki [2], Takeshi Kasaya [2] & Yoshiaki Nakamura [1] ✉

Toward drastic enhancement of thermoelectric power factor, quantum confinement effect proposed by Hicks and Dresselhaus has intrigued a lot of researchers. There has been much effort to increase power factor using step-like density-of-states in two-dimensional electron gas (2DEG) system. Here, we pay attention to another effect caused by confining electrons spatially along one-dimensional direction: multiplied 2DEG effect, where multiple discrete subbands contribute to electrical conduction, resulting in high Seebeck coefficient. The power factor of multiple 2DEG in GaAs reaches the ultrahigh value of ~100 $\mu$Wcm$^{-1}$K$^{-2}$ at 300 K. We evaluate the enhancement rate defined as power factor of 2DEG divided by that of three-dimensional bulk. The experimental enhancement rate relative to the theoretical one of conventional 2DEG reaches anomalously high (~4) in multiple 2DEG compared with those in various conventional 2DEG systems (~1). This proposed methodology for power factor enhancement opens the next era of thermoelectric research.

Human beings have been seeking a powerful solution to the energy crisis. Thermoelectric (TE) material, which enables the direct conversion between waste heat and electricity, is attracting worldwide interests as one of the sustainable power sources[1,2]. The TE performance is quantified by a dimensionless figure-of-merit ZT; ZT = $S^2\sigma T/\kappa$, where $S$ is Seebeck coefficient, $\sigma$ is electrical conductivity, $\kappa$ is thermal conductivity, $T$ is absolute temperature, and $S^2\sigma$ is power factor (PF). The ZT increase has been done by two approaches: $\kappa$ reduction or PF enhancement[3–15]. In 2000s, nanostructuring approach intensified interface phonon scattering, decreasing $\kappa$ drastically. Some studies achieved 100–200 times smaller $\kappa$ by introducing nanostructures, making a big impact on TE research[3–8]. On the other hand, in the ever-reported methodologies of PF enhancement, the enhancement rate of PF is achieved to be several times (2–3 times for energy filtering effect[10–12], 1.5–2 times for resonant scattering effect[13,14]). Epoch-making methodologies for PF enhancement have been expected for further increase in thermoelectric performance.

In 1993, Hicks and Dresselhaus proposed the concept of PF enhancement by quantum confinement effect[16]; e.g. step-like

density of states (DOS) in two-dimensional electron gas (2DEG) system increases $S$ (step-like DOS effect) (Supplementary Note 1). Since then, much effort has been made to experimentally demonstrate PF enhancement by quantum confinement effect[17–22]. In 2018, Zhang et al. experimentally observed an evident feature of 2DEG in SrTiO$_3$[23]: the phenomenon of $S$ enhancement brought by decreasing $t_{2DEG}/\lambda$, where $t_{2DEG}$ is 2DEG channel thickness and $\lambda$ is the thermal de Broglie wavelength[23–25]. Furthermore, PF enhancement has been tried by a combination of step-like DOS effect for high $S$ and modulation doping effect for high carrier mobility $\mu$ (Fig. 1a)[17,21]. Toward further high enhancement rate $R_{2D/3D}$ defined as $R_{2D/3D}$ = PF$_{2DEG}$/PF$_{3D}$, where PF$_{2DEG}$ is PF of 2DEG and PF$_{3D}$ is PF of three-dimensional (3D) bulk, it is strongly demanded to obtain more drastic increase of $R_{2D/3D}$ as a function of $t_{2DEG}/\lambda$ than theoretical function $R_{2D/3D}$ (($R_{2D/3D}$)$_{th}$) reported in the previous study[24] (Fig. 1d). Although step-like DOS effect by quantum confinement has been spotlighted so far, we pay attention to another effect caused by quantum confinement effect: multiple discrete subbands with step-like DOS. Provided that multiple subbands with step-like DOS at higher

[1]Osaka University, 1-3 Machikaneyama-cho, Toyonaka, Osaka 560-8531, Japan. [2]National Institute for Materials Science, 1-2-1 Sengen, Tsukuba, Ibaraki 305-0047, Japan. ✉e-mail: nakamura.yoshiaki.es@osaka-u.ac.jp

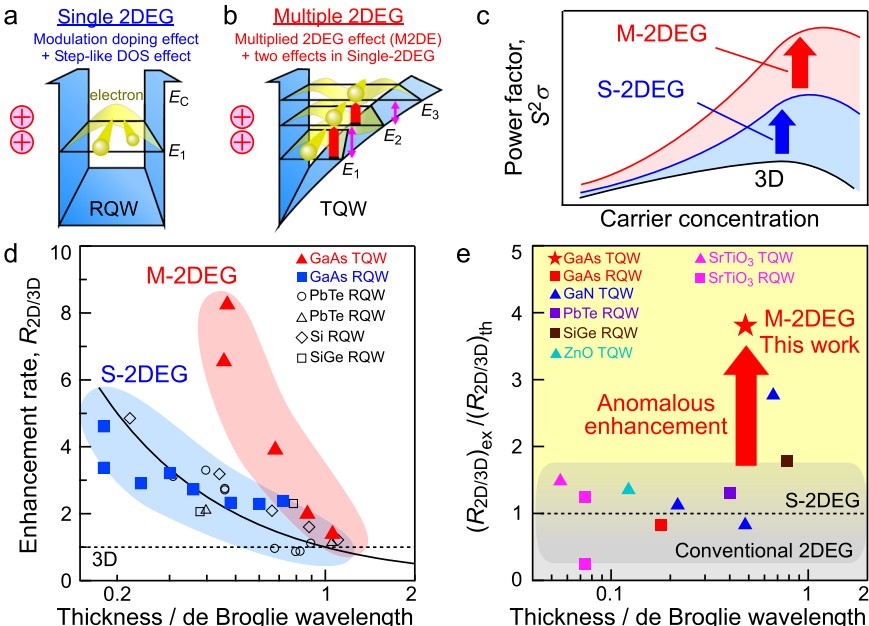

**Fig. 1 | Power factor $S^2\sigma$ enhancement by multiplied 2DEG effect (M2DE).**
**a** Schematic illustration of single 2DEG (S-2DEG) in rectangular quantum well (RQW), where modulation doping effect increases carrier mobility $\mu$ and step-like density-of-states (DOS) effect originated in quantum confinement effect increases Seebeck coefficient $S$. **b** Schematic illustration of multiple 2DEG (M-2DEG) in triangular quantum well (TQW), where M2DE bringing high $S$ appears in addition to modulation doping effect for high $\mu$ and step-like DOS effect for high $S$. **c** Schematic illustration of $S^2\sigma$ enhancement by three effects: modulation doping effect, step-like DOS effect, and M2DE. **d** The enhancement rate of $S^2\sigma$ ($R_{2D/3D}$) as a function of the 2DEG channel thickness/de Broglie wavelength. The solid triangles, the solid squares, and the open marks are $R_{2D/3D}$ values of M-2DEG with M2DE (This study), S-2DEG without M2DE or with almost no M2DE (This study), and 2DEG without

M2DE or with almost no M2DE (Preceding studies by other groups: PbTe RQW[17] (the open circles), PbTe RQW[18] (the open triangles), Si RQW[19] (the open diamonds), SiGe RQW[21] (the open squares)), respectively. The solid line represents the theoretical $R_{2D/3D}$ without M2DE ($R_{2D/3D}$)$_{th}$[24] which is consistent with the data of S-2DEG and preceding data by the other groups. The broken line denotes $R_{2D/3D} = 1$ corresponding to the performance of 3D materials. **e** Experimental $R_{2D/3D}$ ($R_{2D/3D}$)$_{ex}$ divided by theoretical $R_{2D/3D}$ without M2DE ($R_{2D/3D}$)$_{th}$. In this work, GaAs TQW (the red star) and GaAs RQW (the red square). In the preceding results, GaN TQW[30–32] (the blue triangles), PbTe RQW[17] (the purple square), SiGe RQW[21] (the brown square), ZnO TQW[33] (the light blue triangle), SrTiO₃ TQW[20] (the pink triangle), and SrTiO₃ RQW[20,22] (the pink squares).

energy, which are formed by quantum confinement in two-dimensional electron gas (2DEG) systems[26,27], contributed to electrical conduction, $S$ would be substantially enhanced because the participation rate of higher-energy carriers in the carrier conduction becomes larger (Fig. 1b, c); we call multiplied two-dimensional electron gas effect (M2DE). In this study, we choose GaAs as a material to demonstrate M2DE. Therein, quantum confinement effect easily appears because $t_{2DEG}/\lambda$ of GaAs becomes sufficiently small for 2DEG even in relatively large $t_{2DEG}$ due to its relatively long $\lambda$. In addition, GaAs, which is applied to photonic devices such as vertical cavity surface emitting laser for smart phone, is an ideal material in terms of social application.

Here, we demonstrate that M2DE brings drastic PF enhancement as follows. We form GaAs triangular quantum well (TQW) with M2DE in addition to modulation doping effect and step-like DOS effect (Fig. 1b, c)[28]. TQW samples exhibit higher $S$ than rectangular quantum well (RQW) samples without M2DE or with almost no M2DE when comparing $S$ values under the situation that the channel width $t_{ch}$ of TQW is equal to the well width $t_{well}$ of RQW. This indicates that multiple 2DEG (M-2DEG) in TQW with M2DE is more promising than conventional single 2DEG (S-2DEG) in RQW without M2DE. The PF of M-2DEG reaches the maximum value of ~100 µW cm⁻¹ K⁻² at $n$ of ~1×10¹⁸ cm⁻³ at 300 K, which is in a class of ultrahigh PF. Thanks to M2DE, M-2DEG shows more drastic increase of $R_{2D/3D}$ with decreasing $t_{2DEG}/\lambda$ than S-2DEG (Fig. 1d). The experimental $R_{2D/3D}$ (($R_{2D/3D}$)$_{ex}$) relative to the theoretical $R_{2D/3D}$ without M2DE (($R_{2D/3D}$)$_{th}$) is anomalously high in M-2DEG compared with those in various conventional 2DEG systems (~1) (Fig. 1e)[17,20–24,29–33]. Therein, the layered materials are excluded owing to the difficulty in discussing the contribution of

M2DE in the layered materials because the electronic band structure related to the layer number[34,35] influences on the TE properties. This proposed methodology for PF enhancement opens the next era of TE research.

## Results

### Sample structures and calculated energy band diagrams

The TQW and RQW samples were formed for M-2DEG and S-2DEG, respectively, using molecular beam epitaxy (MBE). Illustrations of sample structures and simple band diagrams are shown in Fig. 2a, b, where conduction band bottom of 3D GaAs ($E_c$), carrier energy ($E$) and the bottom energy of $i$-th subband ($E_i$). The index $i$ ($i = 1, 2,...$) is the subband number, where the subband bottom with the smaller number of $i$ positions at the lower energy level. In general RQW, the energy difference between discrete subband bottoms ($E_{i+1}-E_i$) is monotonically increasing with increase in the $i$ value. Therefore, unlike TQW, it is expected that one subband (or two subbands) can only exist in the present AlGaAs/GaAs/AlGaAs RQW with ~0.2 eV barrier height when the step-like DOS appears due to the sufficiently small $t_{well}$, indicating S-2DEG system (Supplementary Note 2). Modulation doping was performed for both samples by inserting Si-doped $Al_{0.3}Ga_{0.7}As$ layers as carrier suppliers. In TQW and RQW, 2DEG channels were formed at the interfaces of undoped GaAs/$Al_{0.3}Ga_{0.7}As$ spacer and in the quantum well of GaAs layers sandwiched by two $Al_{0.3}Ga_{0.7}As$ layers, respectively. In TQW, electron Hall concentration $n$ values of channels were tuned by controlling the thicknesses of spacer layers $t_{sp}$ (0, 2, 30, 60, and 90 nm). The control of $t_{sp}$ also changed the energy band structure[36], bringing the $t_{ch}$ variation from 8 to 18 nm. In RQW, $t_{well}$ was controlled from 3 to 12 nm.

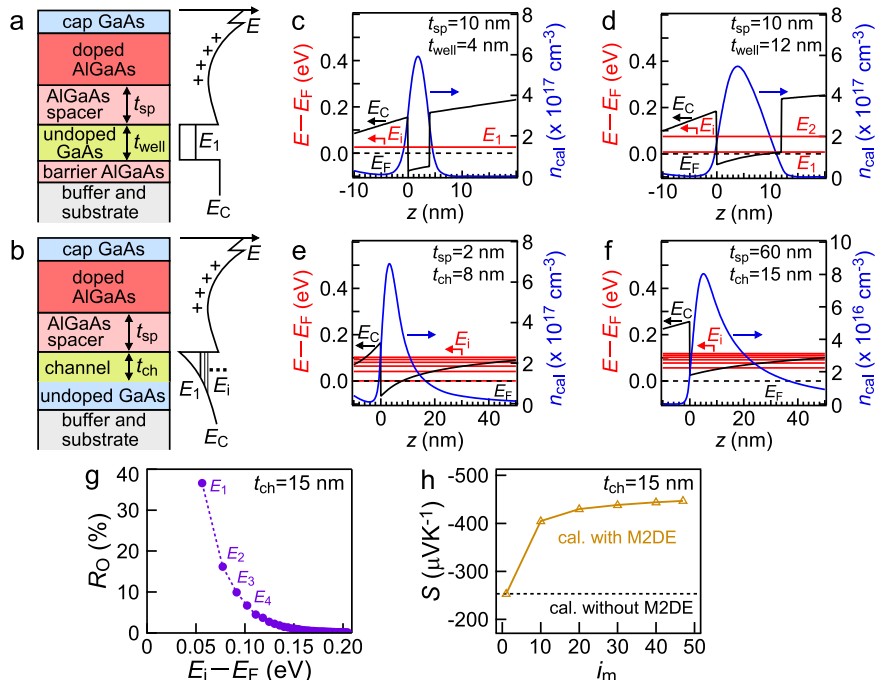

**Fig. 2 | Sample structure illustrations, calculated energy band diagrams, and theoretical demonstration of multiplied 2DEG effect (M2DE). a, b** Illustrations of sample structures and energy band diagrams of rectangular quantum well (RQW) (**a**) and triangular quantum well (TQW) samples (**b**)[48]. **c–f** Calculated energy band diagrams of RQW with the well width $t_{well}$ = 4 nm (**c**), 12 nm (**d**) and TQW samples with the channel width $t_{ch}$ = 8 nm (**e**), 15 nm (**f**). The solid black lines: conduction band bottom of 3D GaAs ($E_c$), the broken black lines: Fermi energy $E_F$, the solid red lines: the bottom energy of $i$-th subband ($E_i$) (for simplicity, $E_i$ with $i < 7$ are displayed), and the solid blue line: the calculated carrier distribution $n_{cal}$ as a function of $z$. $z$ is the distance from the interface of undoped GaAs/AlGaAs spacer along the direction perpendicular to the sample surface. **g** The carrier occupation ratio $R_O$ as a function of $E_i$-$E_F$ in the TQW sample with $t_{ch}$ = 15 nm. **h** Calculated Seebeck coefficient $S$ as a function of $i_m$ in the TQW sample with $t_{ch}$ = 15 nm, when the contribution of the $i$-th subband is considered until $i_m$.

We reveal that multiple subbands can contribute to electrical conduction in TQW, not in RQW. As examples of calculation model, we consider the samples of RQW with $t_{well}$ = 4, 12 nm (Fig. 2c, d), and TQW with $t_{ch}$ = 8, 15 nm (Fig. 2e, f). The energy band diagrams and the calculated carrier distribution, $n_{cal}$ were obtained by self-consistent computation using one-dimensional Poisson-Schrödinger equation[37]. It was found that the TQW is formed at the interface of undoped GaAs/ AlGaAs spacer. Therein, multiple subbands locate near Fermi energy $E_F$. For example, some subbands locate in the range of $E$-$E_F$ < ~ 0.1 eV (Fig. 2e, f) in the TQW unlike only one or two subbands in the RQW (Fig. 2c, d, Supplementary Note 2).

### Theoretical demonstration of M2DE

To clarify the contribution of carrier existing at each subband to electrical conduction, we calculated the occupation ratio $R_O$ defined as $R_O = n_i/n_t$, where $n_i$ is sheet carrier concentration at the $i$-th subband and $n_t$ is the sum of $n_i$. The expressions for $n_i$ and $n_t$ are described as follows:

$$n_i = \int_{E_i}^{\infty} f_0(E) D_i(E) dE \qquad (1)$$

$$n_t = \sum_i n_i \qquad (2)$$

where $f_0(E)$ is the Fermi−Dirac distribution function. $D_i(E)$ is DOS at the $i$-th subband, which is described as $m/\pi\hbar^2$. Therein, $m$ is effective mass of carrier and $\hbar$ is Dirac constant. It is found that $R_O$ is the function of $E_i$ from Eqs. (1) and (2). Figure 2g shows $R_O$ at the $i$-th subband. As $E_i$-$E_F$ increased, $R_O$ decreased nearly exponentially, which is coming from the energy dependence of the Fermi−Dirac distribution function. However, some $R_O$ values at the $i$-th subband ($i > 1$) seem to be

relatively high. This implies that multiple subbands can contribute to electrical conduction[38]. Thus, it is expected that M2DE can appear in TQW.

We theoretically demonstrate $S$ enhancement by M2DE in TQW. As an example of calculation model, we consider the sample with $t_{ch}$ = 15 nm. Theoretical $S$ was calculated under parabolic band for 2DEG and bulk, and relaxation time approximations on the basis of Boltzmann transport theory (details available in Methods). In the summation of $i$-th subband in the calculation, it is enough to consider up to the maximum $i$-th subband contributing to electrical conduction although it is ideal to consider up to infinity. Therefore, when the contribution of the $i$-th subband is summated until $i_m$, we investigated the relationship between $S$ and $i_m$ (Fig. 2h), which was calculated using physical parameters[39,40] displayed in Table 1. $S$ was saturated in the range of $i_m$ > 20 because of less contribution of subbands with $i > 20$ to electrical conduction. This saturation indicates that $i_m$ of 20 is critical value ($i_C$) to calculate $S$ accurately. Namely, it is enough to calculate $S$ using the $i$ of less than $i_C$ (in this case, 20). In this study, calculations of TE properties for TQW samples with various $t_{ch}$ were also performed with $i_m \sim i_C$ for sufficient calculation accuracy (Supplementary Note 3).

### Table 1 | Parameters used in the calculation

| Parameter | Symbol | Value[39,40] |
|---|---|---|
| Effective mass | $m$ | $0.067m_O$ kg |
| Free electron mass | $m_O$ | $9.11 \times 10^{-31}$ kg |
| Relative high-frequency dielectric constant | $\kappa_\infty$ | 10.89 |
| Relative static dielectric constant | $\kappa_O$ | 13.18 |
| Longitudinal optical phonon energy | $\hbar\omega_{LO}$ | 36.5 meV |
| Deformation potential constant | $D_A$ | 13.5 eV |
| Longitudinal elastic constant | $c_L$ | $1.4 \times 10^{11}$ N m$^{-2}$ |

In the sample with $t_{ch}$ = 15 nm (Fig. 2h), the saturated $S$ value in the calculation with multiple subbands ($i_m$ > 20) was ~1.7 times higher than that in the calculation with single subband ($i$ = 1), namely the calculation without M2DE. This theoretically proves that M2DE substantially enhances $S$.

### Thermoelectric properties

Experimental and calculated TE properties of M-2DEG and S-2DEG are shown in Fig. 3. Therein, theoretical calculation of $S$ and $\mu$ was performed under parabolic band and relaxation time approximations on the basis of Boltzmann transport theory[41]. The details of carrier scattering models and used parameters are written in the section of Numerical calculation and Table 1 respectively. Figure 3a, b show $S$ and $\mu$ as a function of $n$ at 300 K, respectively. When estimating $n$ of M-2DEG in TQW, we defined the $t_{ch}$ as FWHM of the carrier concentration distribution along the perpendicular direction to substrate surface (Supplementary Note 4)[20]. In M-2DEG (TQW), $n$ was tuned by controlling $t_{sp}$. As shown in Fig. 3a, when decreasing $t_{sp}$ ($t_{ch}$), $n$ was increased because of increase of carrier supply from Si-doped $Al_{0.3}Ga_{0.7}As$ layers. The $S$ values of M-2DEG (the solid red triangles) were compared with that of 3D GaAs film (the solid black circle) that does not have modulation doping effect, step-like DOS effect, and M2DE. We plotted the calculation curve of 3D GaAs[42] which reproduces the experimental value of 3D GaAs film. When comparing them at the same $n$, M-2DEG exhibited higher $S$ than the calculation curve of 3D GaAs. To demonstrate $S$ enhancement by M2DE experimentally, we measured $S$ values of conventional S-2DEG in RQW samples (the solid blue squares) without M2DE or with almost no M2DE for comparison. When varying $t_{well}$ from 12 to 3 nm, $S$ values of S-2DEG were gradually increased because of step-like DOS effect. This tendency was well reproduced by the $S$ calculation for S-2DEG (the open blue squares). Thus, not only M2DE but also step-like DOS effect causes $S$ enhancement, making it difficult to understand the physical mechanism of $S$ enhancement. To discuss the difference between the two effects, let us compare $S$ values of M-2DEG with those of S-2DEG. At almost the same $n$, the M-2DEG with $t_{ch}$ of ~8 nm exhibited higher $S$ than S-2DEG with $t_{well}$ of ~8 nm, while in the stronger confinement case of small width ($t_{well}$ ~ 3 nm) in RQW, high $S$ was obtained to be comparable to that in the case of 8 nm width in TQW. This is because $S$ enhancement appears in M-2DEG (TQW) over a wide range of confinement width, unlike

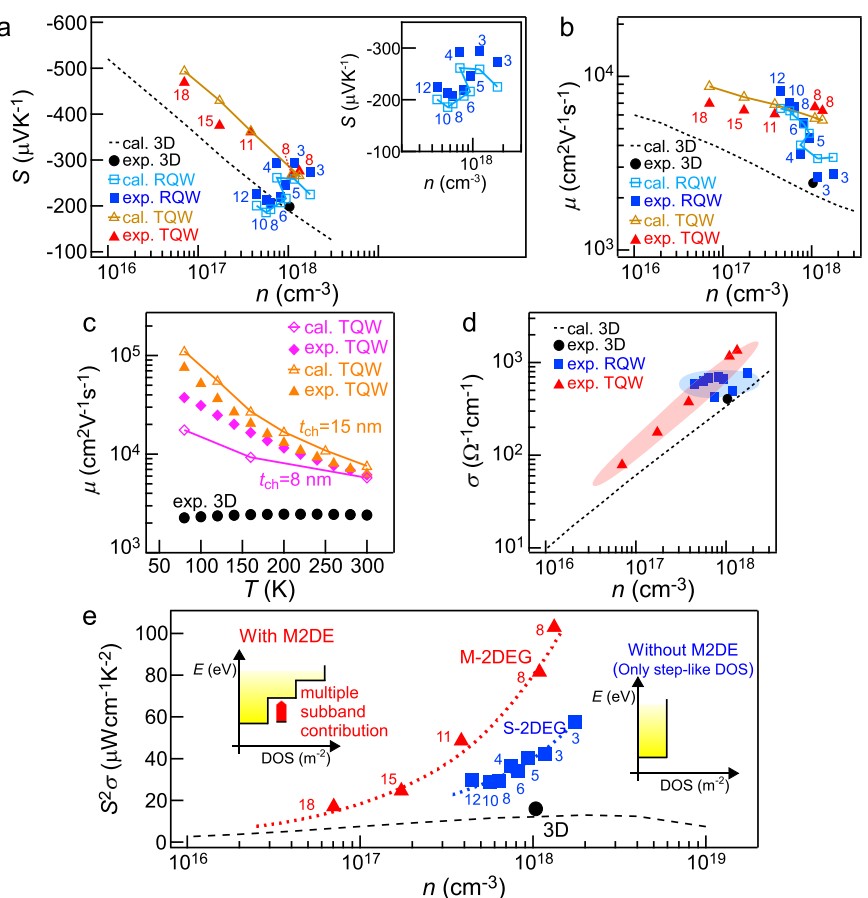

**Fig. 3 | Thermoelectric properties. a, b** Carrier concentration $n$ dependences of Seebeck coefficient $S$ (**a**) and carrier mobility $\mu$ (**b**) measured at 300 K in multiple 2DEG (M-2DEG) in triangular quantum well (TQW) with multiplied 2DEG effect (M2DE) (the solid red triangles), single 2DEG (S-2DEG) in rectangular quantum well (RQW) without M2DE or with almost no M2DE (the solid blue squares), respectively. The calculation data for M-2DEG (the open yellow triangles) and S-2DEG (the open blue squares) are also plotted simultaneously. For comparison with the data in 3D GaAs without 2DEG, the experimental value (the solid black circle) and calculation curves (the broken lines) of 3D GaAs are simultaneously plotted. The channel width $t_{ch}$ of M-2DEG and the well width $t_{well}$ of S-2DEG are displayed around the experimental data points. The inset in (**a**) is an enlarged $n$-$S$ plot: experimental and calculated $n$ dependences of $S$ in S-2DEG. **c** Temperature $T$ dependences of $\mu$ in M-2DEG with $t_{ch}$ = 8 (the solid diamonds) and 15 nm (the solid triangles), 3D GaAs film without 2DEG (the solid circles). We also simultaneously plotted the calculated $T$-$\mu$ curves of M-2DEG with $t_{ch}$ = 8 (the open diamonds) and 15 nm (the open triangles). **d, e** $n$ dependences of electrical conductivity $\sigma$ (**d**) and power factor $S^2\sigma$ (**e**) at 300 K in M-2DEG with M2DE (the solid triangles), S-2DEG without M2DE or with almost no M2DE (the solid squares), respectively. The experimental data (the solid circles) and the calculation curves (the broken lines) of 3D GaAs are simultaneously plotted. The dotted curves in (**e**) are eye-guides for M-2DEG (red) and S-2DEG (blue). In (**e**), the $t_{ch}$ of M-2DEG and $t_{well}$ of S-2DEG are displayed around the experimental data points. The insets show the density of states (DOS) of M-2DEG (with M2DE) and S-2DEG (without M2DE).

S-2DEG (RQW) with strong $t_{well}$ dependence, which is also confirmed by the calculation (Supplementary Note 5). Furthermore, the $S$ calculation (the open yellow triangles) including M2DE in M-2DEG agreed with the experimental $n$-$S$ data (Fig. 3a) and $T$–$S$ data (Supplementary Note 6), which is the theoretical evidence that M2DE appears. Thus, $S$ enhancement by M2DE was demonstrated both experimentally and theoretically.

As well as $S$, $\mu$ values of M-2DEG were compared with the calculation curve of 3D GaAs (Fig. 3b). When comparing them at the same $n$, M-2DEG with modulation doping effect exhibited higher $\mu$ than the calculation curve of 3D GaAs without modulation doping effect which reproduces the experimental value of 3D GaAs film. Furthermore, the experimental $\mu$ data agreed with the $\mu$ calculation including M2DE in addition to modulation doping effect and step-like DOS effect for M-2DEG. We also obtained $T$–$\mu$ data in the $T$ range of 80-300 K (Fig. 3c). The experimental $T$-$\mu$ data of M-2DEG with $t_{ch}$ of 8 nm were compared with those of 3D GaAs, where compared samples had almost the same $n$ of -1 × 10$^{18}$ cm$^{-3}$ at 300 K. Then, $\mu$ values of the M-2DEG drastically increased as $T$ decreased, while $\mu$ of 3D GaAs film did not depend on the $T$. The tendency of experimental data in the M-2DEG was explained by the theoretical $T$–$\mu$ curve of M-2DEG (the open marks in Fig. 3c), where the dominant scattering is polar optical phonon scattering due to the almost no ionized impurity scattering unlike 3D GaAs with ionized impurities. The orders of magnitude higher mobility at low temperature is reported as the result from modulation doping effect[43,44]. These results strongly support that the modulation doping effect, step-like DOS effect, and M2DE appear in M-2DEG.

On the other hand, as shown in Fig. 3b, S-2DEG with modulation doping effect also exhibited higher $\mu$ than the calculation curve of 3D GaAs without modulation doping effect at the same $n$. When varying $t_{well}$ from 12 to 3 nm, $\mu$ values of S-2DEG were monotonically decreased because of increase of interface carrier scattering rate. This tendency was well reproduced by the $\mu$ calculation for S-2DEG. Thus, S-2DEG has a trade-off relationship between $S$ and $\mu$ with respect to $t_{well}$, making it difficult to realize ultrahigh PF. In contrast, high $\mu$ values of M-2DEG did not depend on the $t_{ch}$ within the range of 8–18 nm. Therefore, M2DE is expected along with the high $\mu$ of -6000 cm$^2$V$^{-1}$s$^{-1}$. Namely, M-2DEG has a high potentiality of exhibiting ultrahigh PF by simultaneous enhancement of $S$ and $\mu$.

Figure 3d shows $\sigma$ as a function of $n$ at 300 K. M- and S-2DEG exhibited higher $\sigma$ values than 3D GaAs at almost the same $n$ because of higher $\mu$. As for the $\sigma$ tendency against $n$, there was a significant difference between M- and S-2DEG; $\sigma$ of M-2DEG increased as $n$ increased, while $\sigma$ of S-2DEG did not depend on the $n$. The increasing $\sigma$ tendency of M-2DEG is explained by constant $\mu$ tendency against $n$ (Fig. 3b). On the other hand, constant $\sigma$ tendency of S-2DEG is attributed to the drastically-decreasing $\mu$ tendency against $n$. When $\sigma$ values of M-2DEG with $t_{ch}$ of 8 nm were compared with those of S-2DEG with $t_{well}$ of 3 nm, where both samples exhibited the equivalent $S$ values at almost the same $n$, M-2DEG had approximately 3 times higher $\sigma$ than S-2DEG. This indicates that M-2DEG is more promising than S-2DEG in terms of simultaneous realization of high $S$ and high $\sigma$.

**Experimentally observed anomalous power factor enhancement in M-2DEG**

Figure 3e shows PF as a function of $n$ at 300 K. Both M- and S-2DEG exhibited higher PF than 3D GaAs at almost the same $n$. When decreasing $t_{well}$ from 12 to 3 nm in RQW, PF values of S-2DEG increased monotonically because $S$ was substantially increased by step-like DOS effect. A remarkable fact is that M-2DEG always exhibited much higher PF than S-2DEG because of M2DE. In Fig. 3a, b and e, at the -(1-2) × 10$^{18}$ cm$^{-3}$, higher PF in M-2DEG comes from higher $\mu$ in TQW, where $S$ values of TQW and RQW are comparable, while at -4 × 10$^{17}$ cm$^{-3}$, higher PF in M-2DEG is due to higher $S$ in TQW, where $\mu$ values of TQW and RQW are comparable. This is because there is a trade-off relationship

between $S$ and $\mu$ in RQW. On the other hand, $S$ and $\mu$ are simultaneously enhanced in TQW with M2DE. As a result, the maximum PF of M-2DEG reached -100 μW cm$^{-1}$ K$^{-2}$ at $n$ of -1 × 10$^{18}$ cm$^{-3}$ at 300 K, which is in a class of ultrahigh PF. Thanks to the $S$ enhancement by M2DE along with the high $\mu$, M-2DEG showed more drastic increase of $R_{2D/3D}$ with $t_{2DEG}/\lambda$ decrease than S-2DEG (Fig. 1d). As a result, M-2DEG exhibited the highest $(R_{2D/3D})_{ex}/(R_{2D/3D})_{th}$ among various 2DEG systems (Fig. 1e), which was anomalously high like singularity compared with those in various 2DEG systems (-1). This highlights that M2DE can bring ultrahigh PF beyond conventional 2DEG.

## Discussion

In summary, we demonstrated that M2DE caused by the quantum confinement effect brings drastic PF enhancement. M-2DEG with M2DE in addition to modulation doping effect and step-like DOS effect exhibited higher $S$ than conventional S-2DEG without M2DE or with almost no M2DE over a wide range of confinement width. The PF of M-2DEG reached the maximum value of -100 μW cm$^{-1}$ K$^{-2}$ at $n$ of -1 × 10$^{18}$ cm$^{-3}$ at 300 K, which is in a class of ultrahigh $S^2\sigma$. Thanks to M2DE, M-2DEG exhibited the highest $(R_{2D/3D})_{ex}/(R_{2D/3D})_{th}$ among various conventional 2DEG systems except for the layered materials with the electronic band structure depending on the layer number (Fig. 1e). This value was anomalously high like singularity compared with those in various conventional 2DEG systems. This study presented the methodology enabling the drastic PF enhancement based on quantum confinement effect, which opens the next era of TE research.

## Methods
### Sample preparation
TQW samples were formed using MBE in the following process. To obtain clean surfaces of undoped GaAs(001) substrates, undoped GaAs (300 nm) initial layers were grown on the GaAs substrates. Subsequently, as the buffer layers, GaAs/Al$_{0.3}$Ga$_{0.7}$As superlattice layers were grown on the undoped GaAs (300 nm)/GaAs substrates by alternately depositing GaAs (10 nm) and Al$_{0.3}$Ga$_{0.7}$As (10 nm) 20 times. On the buffer layers, undoped GaAs (1000 nm) layers with high crystallinity were grown. These layers were grown at 893 K. After the growth of Al$_{0.3}$Ga$_{0.7}$As spacer layers on the undoped GaAs (1000 nm) layers at 893 K, Si-doped Al$_{0.3}$Ga$_{0.7}$As (dopant concentration: 5 × 10$^{17}$ cm$^{-3}$, thickness: 80 nm) layers were grown at 823 K to supply carrier to the interface of undoped GaAs (1000 nm)/Al$_{0.3}$Ga$_{0.7}$As spacer. The $n$ was tuned by controlling $t_{sp}$ (0, 2, 30, 60, and 90 nm). Finally, to prevent the oxidation of samples, the sample surfaces were capped by depositing GaAs layers (10 nm) at 823 K.

For reference, RQW samples without M2DE or with almost no M2DE were formed. Undoped GaAs (500 nm) layers were grown on the GaAs(001) substrates. Subsequently, as the buffer layers, GaAs/AlAs superlattice layers were grown on the undoped GaAs (500 nm)/GaAs substrates by alternately depositing GaAs (2 nm) and AlAs (2 nm) 100 times. On the buffer layers, undoped Al$_{0.3}$Ga$_{0.7}$As (20 nm) barrier layers, GaAs (3, 4, 5, 6, 8, 10, and 12 nm) layers, and Al$_{0.3}$Ga$_{0.7}$As (2, 10 nm) spacer layers were grown in a sequential order. The growths of these layers were carried out at 873 K. After that, as the carrier suppliers to GaAs wells, Si-doped Al$_{0.3}$Ga$_{0.7}$As (dopant concentration: 7 × 10$^{17}$ cm$^{-3}$, thickness: 80 nm) layers were grown at 813 K on the Al$_{0.3}$Ga$_{0.7}$As (2 or 10 nm) spacer layers. Finally, to prevent the oxidation of samples, capping GaAs layers (10 nm) were formed at 813 K.

### Thermoelectric property measurements
The stacked structures of AuGe/Ni/Au were formed on the samples as electrodes. To make ohmic contact, the samples were annealed at 723 K for 90 s. Sheet electrical conductivity and sheet carrier concentration were measured using the van der Pauw method and Hall effect measurement, respectively. $\sigma$ and $n$ are obtained by dividing measured sheet electrical conductivity and sheet carrier concentration

by $t_{well}$ or $t_{ch}$[20]. In our Hall effect measurement, we used 2401 source-meter (Keithley) as source measure unit, and the range of magnetic field is from −0.5 T to 0.5 T. Therein, the errors of $n$ and $\mu$ are about 13%. $S$ was measured using ZEM-3 (ADVANCE RIKO Inc.)[45,46], where the temperature difference is applied along the in-plane direction, and the differences of temperatures and the electric voltages between two points on the films were obtained by thermocouple probes. The contribution of Si-doped $Al_{0.3}Ga_{0.7}As$ layer to electrical conduction was removed using the parallel conduction model (Supplementary Note 7).

## Numerical calculation

Theoretical $T-\mu$, $n-\mu$, and $n-S$ curves were calculated under effective mass and relaxation time approximations on the basis of Boltzmann transport theory as follows:

$$S = -\frac{1}{eT}\frac{\sum_i \int_{-\infty}^{\infty}(E-E_i)(E-E_F)\frac{\partial f_0}{\partial E}\tau_i(E-E_i)D_i(E-E_i)dE}{\sum_i \int_{-\infty}^{\infty}(E-E_i)\frac{\partial f_0}{\partial E}\tau_i(E-E_i)D_i(E-E_i)dE} \quad (3)$$

$$\mu = \frac{e}{m}\frac{\sum_i \int_{-\infty}^{\infty}(E-E_i)\frac{\partial f_0}{\partial E}\tau_i(E-E_i)D_i(E-E_i)dE}{\sum_i \int_{-\infty}^{\infty}(E-E_i)\frac{\partial f_0}{\partial E}D_i(E-E_i)dE} \quad (4)$$

where $e$ is the elementary charge and $\tau_i$ is the total carrier relaxation time at the $i$-th subband. $D_i(E)$ was simply assumed as a step function. $1/\tau_i$ is described as the sum of each scattering rate at the $i$-th subband through Matthiessen's rule as follows: $1/\tau_i = 1/\tau_{POP} + 1/\tau_{ADP} + 1/\tau_{RII} + 1/\tau_{IFR}$, where $1/\tau_{POP}$ is polar optical phonon (POP) scattering rate, $1/\tau_{ADP}$ is acoustic deformation potential (ADP) scattering rate, $1/\tau_{RII}$ is remote ionized impurity (RII) scattering rate, $1/\tau_{IFR\_rec}$ is interfacial roughness (IFR) scattering rate in RQW, and $1/\tau_{IFR\_tri}$ is IFR scattering rate in TQW. Inter-subband and intra-subband scatterings are both considered by choosing proper wave functions in POP and ADP scattering calculations. Each scattering rate is described as follows[39,47–50]:

$$\frac{1}{\tau_{POP}} = \frac{e^2 m \hbar \omega_{LO}}{8\pi^2 \hbar^3 \varepsilon_0}\left(\frac{1}{\kappa_\infty}-\frac{1}{\kappa_0}\right)\frac{1}{1-f_0(E)}\sum_j\left([1-f_0(E+\hbar\omega_{LO})]N_q\int\frac{|I(q_z)|^2}{q_+^2+q_z^2}dq_z \right.$$
$$\left. + [1-f_0(E-\hbar\omega_{LO})]u(E-\hbar\omega_{LO})(N_q+1)\int\frac{|I(q_z)|^2}{q_-^2+q_z^2}dq_z\right) \quad (5)$$

$$\frac{1}{\tau_{ADP}} = \frac{D_A^2 k_B T}{2\hbar c_L}\sum_j\left\{\int_{-\infty}^{\infty}\varphi_i(z)\varphi_j(z)dz\right\}D_j(E) \quad (6)$$

$$\frac{1}{\tau_{RII}} = \int_{t_{sp}}^{t_{sp}+t_{dope}}\left[\frac{1}{2}n_{imp}\left(\frac{m}{\pi\hbar^3}\right)\left(\frac{e^2}{2\kappa_0\varepsilon_0}\right)^2\int_0^{2\pi}\frac{\exp(-2q\cdot z)}{(q+q_{TF})^2}(1-\cos\theta)d\theta\right]dz \quad (7)$$

$$\frac{1}{\tau_{IFR\_rec}} = \frac{4\pi m E_i^2 \Delta^2 \Lambda^2}{\left(t+\sqrt{\frac{2\hbar^2}{m(V_0-E_i)}}\right)^2 \hbar^3}\cdot\frac{1}{2\pi}\exp\left(-\frac{\Lambda^2\left(\frac{2m(E-E_i)}{\hbar^2}\right)(1-\cos\theta)}{2}\right)(1-\cos\theta)\frac{q}{q+\frac{2}{a_B}} \quad (8)$$

$$\frac{1}{\tau_{IFR\_tri}} = \frac{m\Delta^2\Lambda^2 e^2\left(e\left(\frac{n}{2}+n_{depl}\right)\right)^2}{2\hbar^3\left(\kappa_0\varepsilon_0+\frac{1}{q}\cdot\frac{e^2 m}{2\pi\hbar^2}\cdot F(q)\right)^2}\exp\left(-\frac{\Lambda^2 q^2}{4}\right)(1-\cos\theta) \quad (9)$$

where $\hbar\omega_{LO}$ is the longitudinal optical phonon energy, $\varepsilon_0$ is the vacuum dielectric constant, $\kappa_\infty$ is the relative high-frequency dielectric constant, $\kappa_0$ is the relative static dielectric constant, $N_q$ is the distribution function of optical phonon. $|I(q_z)|^2$ is the form factor due

to the quantized wave function; $I(q_z) = \int_{-\infty}^{\infty}\varphi_i(z)\varphi_j(z)\exp(iq_z z)dz$, where $\varphi_i(z)$ is the wave function at $i$-th subband, $z$ is the distance along the direction perpendicular to the sample surface ($z=0$ is defined as the interface position of undoped GaAs/AlGaAs spacer.), $q_z$ is the scattering wave vector in the $z$ direction. $\mathbf{q}$ described as $\mathbf{q}=\mathbf{k_2}-\mathbf{k_1}$ is a two-dimensional scattering wave vector from initial state $\mathbf{k_1}$ to the final state $\mathbf{k_2}$ in the elastic collisions. $\mathbf{q_+}$ and $\mathbf{q_-}$ are two-dimensional scattering wave vectors in the phonon absorption and the phonon emission, respectively, as follows:

$$q_+ = \sqrt{2\cdot\left(\frac{2m(E-E_i)}{\hbar^2}\right)+\frac{2m\hbar\omega_{LO}}{\hbar^2}-2\sqrt{\frac{2m(E-E_i)}{\hbar^2}}\cdot\sqrt{\frac{2m(E-E_i)}{\hbar^2}+\frac{2m\hbar\omega_{LO}}{\hbar^2}}\cos\theta} \quad (10)$$

$$q_- = \sqrt{2\cdot\left(\frac{2m(E-E_i)}{\hbar^2}\right)-\frac{2m\hbar\omega_{LO}}{\hbar^2}-2\sqrt{\frac{2m(E-E_i)}{\hbar^2}}\cdot\sqrt{\frac{2m(E-E_i)}{\hbar^2}-\frac{2m\hbar\omega_{LO}}{\hbar^2}}\cos\theta} \quad (11)$$

$$q = 2\sqrt{\frac{2m(E-E_i)}{\hbar^2}}\sin\frac{\theta}{2} \quad (12)$$

where $\theta$ is the scattering angle between $\mathbf{k_1}$ and $\mathbf{k_2}$. $D_A$ is the deformation potential constant, $k_B$ is Boltzmann constant, $c_L$ is the longitudinal elastic constant, $t_{dope}$ is the thickness of Si-doped $Al_{0.3}Ga_{0.7}As$ layer, $n_{imp}$ is the concentration of impurity atoms, $q_{TF}$ is Thomas-Fermi wave number, $V_0$ is the energy barrier height, $\Delta$ and $\Lambda$ are the mean interface roughness values at the $z$ direction and at the perpendicular direction to $z$, respectively (in this calculation, these parameters are fixed at $\Delta=0.5$ nm and $\Lambda=5$ nm), $a_B$ is the effective Bohr radius, $n_{depl}$ is the charge density of the depletion layer, and $F(q) = \int dz\int dz'|\varphi(z)|^2|\varphi(z')|^2\exp(-q|z-z'|)$. In the calculation, $m$ value shown in Table 1 was used for each subband under the assumption that the non-parabolicity effect on $m$ is negligible[51].

The energy band diagram was computed using 1D Poisson solver developed by G. Snider; wavefunction and carrier concentration distribution were self-consistently computed using the Poisson–Schrodinger equation. This computation revealed the level of discrete subband bottom energy, which was used for the calculation of theoretical curves.

The definition of $t_{ch}$ in TQW is essential for estimating TE performance. In this study, we estimated $t_{ch}$ from the carrier distribution profile along the perpendicular direction to the substrate surface; we defined FWHM of the carrier concentration distribution as $t_{ch}$ (Supplementary Note 4).

## Data availability

The authors declare that the data supporting the findings of this study are available within the paper and its supplementary information files, and the data that support the findings of this study are available from the corresponding author upon reasonable request.

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

## Acknowledgements

This work was supported by Grant-in-Aid for Scientific Research A (Grant No. 19H00853, and 23H00258) (Y.N.) and JSPS Fellows (T22KJ2052) (Y.U.). We would like to thank Dr. T. Noda (NIMS) for some discussion.

## Author contributions

Y.N. conceived the idea. Y.U. measured thermoelectric properties of the samples. Y.U. and T.I. performed numerical calculation. Y.U., T.I., and Y.N. discussed the physics and wrote the paper. Y.N. is the principal investigator of this work. T.M. and A.O. formed the samples. H.T.M. and T.K. formed the electrode. All authors discussed the results and contributed to the revision of the final manuscript.

## Competing interests

The authors declare no competing interests.
