## [Peer Review File · Nature Communications]

REVIEWER COMMENTS

Reviewer #1 (Remarks to the Author):

This manuscript demonstrates that the one-sided quantum wells showing triangular band diagram (as the authors call it TQW) can enhance the thermoelectric power factor over those of the double-sided quantum wells of rectangular quantum wells (RQW), and also beyond the bulk limit. This is interesting and as far as I know, this is a new result on the power factors of one-sided quantum wells. However, I have some doubts on the authors' analysis and interpretation of the results, and also have some suggestions on the following few points.

1) As discussed in Line 142 ~ 148, I agree that both M-2DEG and S-2DEG show enhanced Seebeck coefficient over the bulk. However, I don't agree that M-2DEG necessarily shows higher Seebeck coefficient than those of S-2DEG. The authors compared these two at the same channel width, but the channel width for TQW is defined as nothing but the FWHM of the differential carrier concentration over a large distance (n_{cal} in Fig. S4), while the channel width for RQW is the actual width (thickness) of the channel layer. In TQW, carriers go deep in the active layer and there is still significant contribution from those carriers dispersed beyond the "effective" channel width on the Seebeck coefficient. Therefore, it may not be a fair comparison if done at the same effective channel width. Instead, I think that a better comparison can be done at the same volumetric carrier concentration, which is already displayed in Fig. 3. At the same volumetric carrier concentration, the Seebeck coefficient of RQW is similar or even higher than that of TQW above 10^{18} cm^{-3} ! The large enhancement of S in RQW at high carrier concentrations may be due to the increased $\text{abs}(E_1 - E_F)$ with decreasing channel width, which proves the Hicks and Dresselhaus's point of sharp DOS effect. In TQW, however, this effect is small as the higher subbands are closely located. Instead, the first subband E_1 is very narrow in space with a limited number of k 's available, so carriers tend to occupy higher subbands well above the Fermi level. As a result, higher-energy carriers can participate more in the conduction, which is known to be increasing the Seebeck coefficient like a continuous energy filtering effect. This is happening over the entire carrier concentration range ($10^{16} \sim 10^{18} \text{ cm}^{-3}$), so that the Seebeck is enhanced over the bulk limit.

Back to the comparison between TQW and RQW, the power factor enhancement by TQW over RQW at the same volumetric carrier concentration is apparently due to the enhanced mobility and thus enhanced electrical conductivity, not from the Seebeck coefficient. (Fig. 3d and e) This is due to the less carrier scattering off the walls (double-sided for RQW vs. single-sided for TQW) as the authors already pointed out.

I suggest the authors add clarifications on this matter.

2) In Fig. 1a and b, it will be helpful to show the stacked layer structures, i.e., Figs. S2a and S3a (or simpler versions), for both RQW and TQW next to their band diagrams. From this figure, the readers might like to see how the authors made the TQW vs. RQW without reading the full text.

3) I don't think Fig. 2b and c are necessary. They can be put in Supplementary Information. Also I don't think Eq. (2) and (3) are necessary in the main text either. Instead, some of the band diagrams (if not all of them) shown in Fig. S2 and Fig. S4 might be useful to be displayed in the main text as part of Fig. 2. That way, it'll be clearer how the channel width is controlled – by changing the actual channel layer width for RQW, while by changing the spacer width for TQW. In Fig. S4, it'll also be informative to draw the subband energy lines and show how their distances change with the effective channel width.

4) Overall, extensive English editing might be necessary.

I have a few additional minor comments as below:

1. Line 41-42: \sim a factor of 2 enhancement is still significant. Saying "there is no prospect for the development of TE research" goes too far I think.
2. Line 47: add the reference number for Ohta et al.
3. Line 63: I suggest not to use those assertions that cannot be proved properly such as ", which nobody has paid attention to".
4. Line 71: "with t_{2DEG}/λ decrease" should better read "with decreasing t_{2DEG}/λ ".
5. In Fig. 1e and Line 72, how do you define the "theoretical enhancement factor"?
6. Line 99, 104, 105 and throughout the manuscript: "ith" -> "i-th"
7. Line 113: parabolic band for 2DEG? Or is the parabolic band used for bulk only?
8. In Fig. 3c, does t_c mean " t_{ch} "?

Reviewer #2 (Remarks to the Author):

This work realized an anomalous enhancement of thermoelectric power factor via multiplied two-dimensional electron gas effect. This work is of potential interest for power factor enhancement in thermoelectrics. I agree with the authors that strategies that can enhance the power factor are crucial for the future development of TE research. Therefore, I think this work is novel and might inspire the exploration of new strategies for the development of good TE materials. However, one major problem of this work is that it is really hard to read the manuscript, particularly, considering most of the TE researchers are focusing on bulk materials. Since this is a work dealing with the MBE thin films and discussing the quantum refinement effect, there is a lack of elaboration on the used terms. Moreover, the authors used ZEM-3 to measure the Seebeck coefficient of the thin films, however, the reliability might incur doubts. In addition, there are some major problems that need to be solved. Without addressing these problems, I cannot recommend it for publication in Nature Communications.

1. Why structure in Figure S3a can form a triangular quantum well?
2. The definitions of the subbands and EC in Figure 2 and Figure S3 are unclear.
3. In view of that S-2DEG in RQW with twell of ~ 8 nm does not show S enhancement, is there an SDE effect in this sample?
4. Is it reasonable to assume that there is no contribution of ionized impurity scattering by the fact that the mobility does not vary with carrier concentration?
5. An elaborate description of the measurements is recommended, including the direction of the temperature gradient when measuring the Seebeck coefficient, etc.
6. The measured properties taken as examples in S3 are divided by unit area. How do you get the carrier concentration, S , and carrier conductivity per unit volume?
7. The mechanism behind the Seebeck boost by M2DEG is not elucidated in this work.
8. One technical problem is that the authors used many abbreviations in the main text, which unfortunately reduces the readability of this manuscript.
9. In addition, there is a lack of definitions of the used terms and abbreviations, for example, TQW and RQW refer to triangular quantum well and rectangular quantum well, respectively. An explanation of which quantum well is triangular or rectangular in the beginning they appear will enhance the readability of this manuscript.
10. "Some studies achieved 100-200 times reduction of κ , making a big impact on TE research". This sentence might incur a misunderstanding.

Reviewer #3 (Remarks to the Author):

The manuscript presents an interesting approach with promising results, but several key aspects need to be addressed and clarified before considering acceptance. Specifically, the authors should correct writing issues, provide a review of M2DE, clarify the formation of M-2DEG and S-2DEG, provide

additional experimental evidence, and include missing details on Hall measurement, ionized impurity scattering, and simulation assumptions. Detailed comments are as follows:

1. Writing Issues: There are several writing issues that need to be corrected:

a. In equations 1 and 2, "where E_i is the subband energy at the i th band" is mentioned, but there is no E_i in either equation. Do the authors mean that 'E' in equation 1 is the subband energy at the i th band?

b. Some descriptions in the manuscript are not scientific, for example, "In RQW, the energy difference between discrete subbands at higher energy is quite larger than that between the lowest level in RQW and conduction band bottom of 3D GaAs." The term "quite larger" is vague. Authors should provide a quantitative description, such as "** eV higher than ***".

2. Lack of Review on M2DE: The authors claim the presence of the "multiplied two-dimensional electron gas effect (M2DE)". This appears to be based on quantum confinement and the formation of subbands in two-dimensional electron gas (2DEG) systems, which are well-studied phenomena. However, as a major novelty claimed, there is no review about M2DE in the introduction. A brief description of M2DE with references is necessary.

3. Clarification on M-2DEG and S-2DEG Formation: The authors claim the formation of M-2DEG and S-2DEG in TQW and RQW samples, respectively. It is suggested to highlight the structure difference of these two samples by providing a figure.

4. Lack of Experimental Evidence: In figure 2a, the authors claim the distribution of multiple subbands with a gap around 0.02 eV. This claim, if true, may lead to the enhancement of the Seebeck coefficient at 300K (corresponding to $k_B T$ of 25.9 meV). However, apart from the measured Seebeck at 300K, there is a lack of other experimental evidence to support the claim. A temperature dependence measurement of the Seebeck coefficient would be more convincing to prove the presence of M-2DEG. This measurement offers modulation of the responding energy range of around $2 k_B T$, which is sensitive enough for multiple subbands with a gap around 0.02 eV.

5. Missing Details on Hall Measurement: Details about the Hall measurement are missing. The authors claim the measurement of samples with relatively low carrier concentration of 10^{17} to 10^{18} cm⁻³, which is challenging. Information about the capability of the instrument (e.g., magnetic field, SMU information) is necessary. Also, the error should be stated somewhere, for example, in the legend of figure 3.

6. Lack of Evidence on Absence of Ionized Impurity Scattering: The authors state that "Interestingly, μ values of M-2DEG hardly depended on the n , while the calculation μ curve of 3D GaAs exhibited a monotonic decrease as n increased. This indicates that M-2DEG has no contribution of ionized impurity scattering inside the 2DEG channel because of no impurity in the channel by MDE." The authors must provide solid experimental evidence to prove the absence of ionized impurity scattering. If the M-2DEG effect is present, the measured mobility is an average mobility of multiple bands, which may have varied effective masses, thus causing the observed effect in mobility. Have the authors considered this possibility, and how did they exclude it?

7. Missing Simulation Details: The authors state that "The tendency of experimental data in the M-2DEG was well explained by the theoretical T - μ curve." However, the details of the simulation are missing. For example, what are the assumptions for the scattering mechanism of the plotted theoretical T - μ curve?

Reviewer comments and Point-by-point response

Reviewer #1:

Comment: *This manuscript demonstrates that the one-sided quantum wells showing triangular band diagram (as the authors call it TQW) can enhance the thermoelectric power factor over those of the double-sided quantum wells of rectangular quantum wells (RQW), and also beyond the bulk limit. This is interesting and as far as I know, this is a new result on the power factors of one-sided quantum wells. However, I have some doubts on the authors' analysis and interpretation of the results, and also have some suggestions on the following few points.*

Answer to the comment: Thank you for appreciating your comment that our work is interesting and shows a new result. We are trying to clear the doubts the reviewer pointed out as faithful as possible by answering the following reviewer's comments.

Comment 1-1: *As discussed in Line 142 ~ 148, I agree that both M-2DEG and S-2DEG show enhanced Seebeck coefficient over the bulk. However, I don't agree that M-2DEG necessarily shows higher Seebeck coefficient than those of S-2DEG. The authors compared these two at the same channel width, but the channel width for TQW is defined as nothing but the FWHM of the differential carrier concentration over a large distance (n_{cal} in Fig. S4), while the channel width for RQW is the actual width (thickness) of the channel layer. In TQW, carriers go deep in the active layer and there is still significant contribution from those carriers dispersed beyond the "effective" channel width on the Seebeck coefficient. Therefore, it may not be a fair comparison if done at the same effective channel width. Instead, I think that a better comparison can be done at the same volumetric carrier concentration, which is already displayed in Fig. 3. At the same volumetric carrier concentration, the Seebeck coefficient of RQW is similar or even higher than that of TQW above 10^{18} cm^{-3} !*

Comment 1-2: *The large enhancement of S in RQW at high carrier concentrations may be due to the increased $\text{abs}(E_1 - E_F)$ with decreasing channel width, which proves the Hicks and Dresselhaus's point of sharp DOS effect. In TQW, however, this effect is small as the higher subbands are closely located. Instead, the first subband E_1 is very narrow in space with a limited number of k 's available, so carriers tend to occupy higher subbands well above the Fermi level. As a result, higher-energy carriers can participate more in the conduction, which is known to be increasing the Seebeck coefficient like a continuous energy filtering effect. This is happening over the entire carrier concentration range ($10^{16} \sim 10^{18} \text{ cm}^{-3}$), so that the Seebeck is enhanced*

over the bulk limit.

Comment 1-3: *Back to the comparison between TQW and RQW, the power factor enhancement by TQW over RQW at the same volumetric carrier concentration is apparently due to the enhanced mobility and thus enhanced electrical conductivity, not from the Seebeck coefficient. (Fig. 3d and e) This is due to the less carrier scattering off the walls (double-sided for RQW vs. single-sided for TQW) as the authors already pointed out.*

I suggest the authors add clarifications on this matter.

Answer to the comment 1-2: We agree with the interpretation of step-like DOS effect and M2DE the reviewer mentioned. Especially, the explanation about M2DE can help readers understand because M2DE is the novelty of this manuscript. As the reviewer pointed out, M2DE is caused by the relatively large participation rate of higher-energy carriers in the carrier conduction like the energy filtering effect. In energy filtering effect, S enhancement caused by the transport control where carrier transport with low energy is prevented in the barrier, while M2DE is brought by multiple discrete subbands at the higher energy.

In page 4, lines 8 ~ 12, we added the above thing; “Provided that multiple subbands with step-like DOS at higher energy, which are formed by quantum confinement in two-dimensional electron gas (2DEG) systems^{26,27}, contributed to electrical conduction, S would be substantially enhanced because **the participation rate of higher-energy carriers in the carrier conduction becomes larger** (Fig. 1b and 1c);”.

In supplementary information, we also added the section (Supplementary Note 5). In this section, we added the explanation of the mechanism; “**M2DE is caused by the relatively large participation rate of higher-energy carriers in the carrier conduction.**”

Answer to the comments 1-1 and 1-3: We agree that M-2DEG GaAs doesn't necessarily show higher Seebeck coefficient than those of S-2DEG GaAs, and that narrow RQW ($t_{\text{well}} \sim 3$ nm) shows similar Seebeck coefficient to that of TQW ($t_{\text{ch}} \sim 8$ nm) at the same carrier concentration.

As the reviewer mentioned, it is important to clarify the difference of M2DE and step-like DOS effect in more detail. So, let's see the dependence of the Seebeck coefficient enhancement on the confinement width at the same carrier concentration because this effect depends on the confinement strength. (As the reviewer pointed out, the same carrier concentration is important.)

As the reviewer pointed out, t_{well} and t_{ch} cannot be equal, but it is very important to understand the effectiveness of the Seebeck coefficient enhancement. Therefore, we performed additional calculation experiments this time, and we calculated the Seebeck coefficient of TQW and RQW with various confinement widths at the same carrier concentration, as shown in Fig. A1. In Fig. A1a, t_{well} or t_{ch} are used as confinement width. But, t_{well} and t_{ch} were defined differently from well structure and carrier distribution broadening, respectively. On the other hand, in Fig. A1b, the same confinement width definition in TQW and RQW are chosen for fair comparison; namely, carrier distribution broadenings in TQW and RQW ($t_{\text{br}} = \text{full width at half maximum of } n_{\text{cal}}$) are used as confinement width. Figure A1a, A1b show that Seebeck coefficient of TQW is larger than that of RQW over a wide range of the width even if in Fig. A1a, the actual channel width for TQW could be larger due to the obscure width definition in triangle well. On the other hand, in very small width, the Seebeck coefficient values of RQW are almost the same as those of TQW, which is the thing the reviewer pointed out. Namely, as for Seebeck coefficient, (1) S enhancement of TQW is larger than that of RQW over a wide range of the width, (2) **in the case of strong confinement**, as the reviewer mentioned, S values of TQW and RQW with t_{well} of <3 nm (Fig. A1a) or t_{br} of <3.4 nm (Fig. A1b) are compatible in the case of 2DEG-GaAs. As for the power factor, TQW (M-2DEG) always shows larger power factor than RQW (S-2DEG) at the same carrier concentration (Fig. 3e in main text). At the $\sim 1 \times 10^{18} \text{ cm}^{-3}$, the reason is larger μ in TQW where S value is comparable between TQW and RQW (Fig. 3a and 3b in main text), while at $\sim 4 \times 10^{17} \text{ cm}^{-3}$, the reason is larger S in TQW where μ value is comparable between TQW and RQW (Fig. 3a and 3b in main text). This is because S and μ are simultaneously enhanced in TQW with M2DE, and because there is a trade-off relationship between S and μ in RQW. What the reviewer pointed out is the thing of TQW and RQW **with strong confinement** at the $\sim 1 \times 10^{18} \text{ cm}^{-3}$. As the reviewer mentioned, in the revised manuscript, we discuss all the above things.

Fig. A1

In page 10, lines 2 ~ 9, we revised the sentences of “To discuss the **difference of the two effects, let us compare S values of M-2DEG with those of S-2DEG. At almost the same n , the M-2DEG with t_{ch} of ~ 8 nm exhibited higher S than S-2DEG with t_{well} of ~ 8 nm, while in the stronger confinement case of small width ($t_{\text{well}} \sim 3$ nm) in RQW, high S was obtained to be comparable to that in the case of 8 nm width in TQW. This is because S enhancement appears in M-2DEG (TQW) over a wide range of confinement width, unlike S-2DEG (RQW) with strong t_{well} dependence, which is also confirmed by the calculation (Supplementary Note 5).”**

In page 12, lines 1 ~ 3, we added the sentence and words of “Therefore, M2DE is expected along with the high μ of $\sim 6000 \text{ cm}^2\text{V}^{-1}\text{s}^{-1}$. Namely, M-2DEG has a high potentiality of exhibiting ultrahigh PF by S and μ enhancement effects.”

In page 13, lines 2 ~ 7, we added the sentences and words of “At the $\sim 1 \times 10^{18} \text{ cm}^{-3}$, higher PF in M-2DEG comes from higher μ in TQW, where S values of TQW and RQW are comparable, while at $\sim 4 \times 10^{17} \text{ cm}^{-3}$, higher PF is due to higher S in TQW, where μ values of TQW and RQW are comparable (Fig. 3a and Fig. 3b). This is because there is a trade-off relationship between S and μ in RQW. On the other hand, S and μ are simultaneously enhanced in TQW with M2DE. As a result, the maximum PF of M-2DEG reached $\sim 100 \mu\text{Wcm}^{-1}\text{K}^{-2}$ at n of $\sim 1 \times 10^{18} \text{ cm}^{-3}$ at 300 K, which is in a class of ultrahigh PF.”

In supplementary information, we added the section (Supplementary Note 5) of “Calculation of confinement width dependence of the Seebeck coefficient at the same carrier concentration.” In this section, we added the explanation of the mechanism of M2DE; “Mechanism of the step-like DOS effect is reported by Hicks and Dresselhaus, which leads to the strong confinement width dependence. On the other hand, M2DE is caused by the relatively large participation rate of higher-energy carriers in the carrier conduction. This depends on the subband number and E_i , leading to the relatively low confinement width dependence.”

Comment 2: In Fig. 1a and b, it will be helpful to show the stacked layer structures, i.e., Figs. S2a and S3a (or simpler versions), for both RQW and TQW next to their band diagrams. From this figure, the readers might like to see how the authors made the TQW vs. RQW without reading the full text.

Answer to the comment 2: We agree that the structures and the band diagrams of RQW and TQW will be helpful for understanding the difference between RQW and TQW in main text. We added the structures and the band diagrams (Fig. A2a, b) to Fig. 2 in main text.

Fig. A2

In Fig. 2a and 2b, simpler versions of stacked layered structures of RQW and TQW with their band diagrams were shown, and the caption was revised as “a, b Illustrations of sample structures and energy band diagrams of RQW (a) and TQW samples (b).”

In page 6, lines 1 ~ 4, we added the sentence of “Illustrations of sample structures and simple band diagrams are shown in Fig. 2a, and 2b, where conduction band bottom of 3D GaAs (E_c), carrier energy (E) and the bottom energy of i -th subband (E_i). The index i ($i=1, 2, \dots$) is the subband number, where the subband bottom with the smaller number of i positions at the lower energy level.”

Comment 3: *I don't think Fig. 2b and c are necessary. They can be put in Supplementary Information. Also I don't think Eq. (2) and (3) are necessary in the main text either. Instead, some of the band diagrams (if not all of them) shown in Fig. S2 and Fig. S4 might be useful to be displayed in the main text as part of Fig. 2. That way, it'll be clearer how the channel width is controlled - by changing the actual channel layer width for RQW, while by changing the spacer width for TQW. In Fig. S4, it'll also be informative to draw the subband energy lines and show how their distances change with the effective channel width.*

Answer to the comment 3: We agree that S2 and S4 might support the reader to understand the properties of channel, so we show them in Fig. 2 in main text as shown in Fig. A2. On the other hand, we think that the existence of Fig. 2b, 2c and Eq. 2, 3 in main text is no problematic as shown in the above Fig. 2.

In Fig. 2, we added four band diagrams shown in Fig. S2 and S4, and slightly modified annotation in Fig. 2h to understand easily. And the caption was revised as **“c, d, e, f Calculated energy band diagrams of RQW with $t_{\text{well}}=4$ nm (c), 12 nm (d) and TQW samples with $t_{\text{ch}}=8$ nm (e), 15 nm (f). The solid black lines: E_c , the broken black lines: E_F , the solid red lines: E_i (for simplicity, E_i with $i<7$ are displayed), and the solid blue line: the calculated carrier distribution, n_{cal} as a function of z . z is the distance from the interface of undoped GaAs/AlGaAs spacer along the direction perpendicular to the sample surface. g R_0 as a function of E_i-E_F in the TQW sample with $t_{\text{ch}}=15$ nm. h Calculated S as a function of i_m in the TQW sample with $t_{\text{ch}}=15$ nm.”**

In page 6, line 16 ~ page 7, line 2, we added the words of “We reveal that multiple subbands can contribute to electrical conduction in TQW, **not in RQW**. As examples of calculation model, we consider **the samples of RQW with $t_{\text{well}}=4, 12$ nm (Fig. 2c, 2d), and TQW with $t_{\text{ch}}=8, 15$ nm (Fig. 2e, 2f). The energy band diagrams and the calculated carrier distribution, n_{cal} were obtained** by self-consistent computation using one-dimensional Poisson-Schrödinger equation³⁷.”

Comment 4: *Overall, extensive English editing might be necessary.*

I have a few additional minor comments as below:

1. *Line 41-42: ~ a factor of 2 enhancement is still significant. Saying “there is no prospect for the development of TE research” goes too far I think.*

Answer to the comment 4.1: In page 3, lines 9 ~ 12, we revised the sentence of “On the other hand, in the ever-reported methodologies of **PF enhancement**, the enhancement rate of PF is **achieved to be** several times (2-3 times for energy filtering effect¹⁰⁻¹², 1.5-2 times for resonant scattering effect^{13,14}). **Epoch-making methodologies for PF enhancement have been expected for further increase in thermoelectric performance.**”

2. Line 47: add the reference number for Ohta et al.

Answer to the comment 4.2: In page 3, line 17, we added the reference number [23], which was referred in the page 4, line 1 of previous manuscript, and we revised the word of “Ohta et al.” to “**Zhang et al.**” because Ohta is the corresponding author and Zhang is the first author of the reference paper.

[23] Zhang, Y. et al. *Nat. Commun.* **9**, 2224 (2018).

3. Line 63: I suggest not to use those assertions that cannot be proved properly such as “, which nobody has paid attention to”.

Answer to the comment 4.3: In page 4, line 18, we removed “, which nobody has paid attention to.”

4. Line 71: “with t_{2DEG}/λ decrease” should better read “with decreasing t_{2DEG}/λ ”.

Answer to the comment 4.4: In page 5, line 8, we revised “with t_{2DEG}/λ decrease” to “**with decreasing t_{2DEG}/λ .**”

5. In Fig. 1e and Line 72, how do you define the “theoretical enhancement factor”?

Answer to the comment 4.5: “theoretical enhancement factor” is defined as the calculated power factor of 2D divided by that of 3D which is reported in the previous study [*Phys. Rev. Lett.* **117**, 036602 (2016).], so we added this description.

[24] *Phys. Rev. Lett.* **117**, 036602 (2016).

In page 4, lines 5 ~ 6, we revised the sentence of “to obtain more drastic increase of $R_{2D/3D}$ as a

function of $t_{2\text{DEG}}/\lambda$ than theoretical function $R_{2\text{D}/3\text{D}}$ ($(R_{2\text{D}/3\text{D}})_{\text{th}}$) reported in the previous study [24] (Fig. 1d).”

6. Line 99, 104, 105 and throughout the manuscript: “ith” -> “i-th”

Answer to the comment 4.6: In the manuscript, we revised “ith” to “i-th.”

7. Line 113: parabolic band for 2DEG? Or is the parabolic band used for bulk only?

Answer to the comment 4.7: Energy bands of both 2DEG and bulk are assumed as parabolic band.

In page 8, line 4, we added “for 2DEG and bulk.”

8. In Fig. 3c, does t_c mean “ t_{ch} ”?

Answer to the comment 4.8: In Fig. 3c, we revised “ t_c ” to “ t_{ch} .”

Fig. A3

In addition, we revised some words to improve English grammar and to increase readability for readers. Revised words are shown with green highlight. Furthermore, description of the citation of supplementary information, subheadings, and abstract were revised based on the formatting instructions, we revised some words, which were marked by blue light highlight. The revisions written in the above-mentioned “answer to the comments” were marked by yellow highlighted.

Reviewer #2:

Comment: *This work realized an anomalous enhancement of thermoelectric power factor via multiplied two-dimensional electron gas effect. This work is of potential interest for power factor enhancement in thermoelectrics. I agree with the authors that strategies that can enhance the power factor are crucial for the future development of TE research. Therefore, I think this work is novel and might inspire the exploration of new strategies for the development of good TE materials.*

Answer to the comment: Thank you for appreciating our work's potential interest, our strategy, and the novelty of this work.

Comment: *However, one major problem of this work is that it is really hard to read the manuscript, particularly, considering most of the TE researchers are focusing on bulk materials. Since this is a work dealing with the MBE thin films and discussing the quantum refinement effect, there is a lack of elaboration on the used terms. Moreover, the authors used ZEM-3 to measure the Seebeck coefficient of the thin films, however, the reliability might incur doubts. In addition, there are some major problems that need to be solved. Without addressing these problems, I cannot recommend it for publication in Nature Communications.*

Answer to the comment: Thank you for your comment. We are trying to address the problems the reviewer pointed out as faithful as possible by answering the following reviewer's comments.

Comment 1: *Why structure in Figure S3a can form a triangular quantum well?*

Answer to the comment 1: A triangular quantum well is formed by Coulomb force of positively charged dopants in 80 nm doped layer of $\text{Al}_{0.3}\text{Ga}_{0.7}\text{As}$ where carriers moved to GaAs channel layer through the spacer layer with t_{sp} thickness as shown below (Fig. A4b). Therein, triangular quantum well potential is composed of the conduction band offset at the interface of $\text{Al}_{0.3}\text{Ga}_{0.7}\text{As}$ /channel layers and linearly-increasing conduction band minimum (E_c). For readers to understand a formation of triangular quantum well in samples, we added the figure of sample structures and simple band diagrams in Fig. 2 in main text as shown below (Fig. A4). and furthermore, we added a famous reference about a triangular quantum well [*Jpn. J. Appl. Phys.* **19**, L225 (1980)].

[28] *Jpn. J. Appl. Phys.* **19**, L225 (1980).

Fig. A4

In Fig. 2, we added structures and band diagrams of the triangular quantum well potential.

In page 6, lines 1 ~ 4, we added the sentence of “**Illustrations of sample structures and simple band diagrams are shown in Fig. 2a, and 2b, where conduction band bottom of 3D GaAs (E_c), carrier energy (E) and the bottom energy of i -th subband (E_i). The index i ($i=1, 2, \dots$) is the subband number, where the subband bottom with the smaller number of i positions at the lower energy level.**”

In page 5, line 2, we added a **reference** about triangular quantum well formation [28].

Comment 2: *The definitions of the subbands and EC in Figure 2 and Figure S3 are unclear.*

Answer to the comment 2: E_c has been defined as energy at the conduction band minimum in caption of Fig. 2, but as the authors pointed out, in the revised manuscript, I defined it in main text, not in the caption.

“Subband” is the energy band in two-dimensional carrier gas system, where the bottom energy of each band is discrete energy level caused by quantum confinement at the z direction. The word of “subband” in quantum well is historically known in the semiconductor research field, but as the author pointed out, we added the brief explanation and some references among the famous literatures [a review paper: *Rev. Mod. Phys.* **54**, 437 (1982)., a text book: Quantum Semiconductor Structures: Fundamentals and Applications (1991). and a regular paper: *Phys. Rev. B* **53**, R10493 (1996).]. We chose two references: [*Rev. Mod. Phys.* **54**, 437 (1982)., *Phys. Rev. B* **53**, R10493 (1996).].

[26] *Rev. Mod. Phys.* **54**, 437 (1982).

[27] *Phys. Rev. B* **53**, R10493 (1996).

In page 6, line 2, we defined E_c as conduction band bottom of 3D GaAs.

In page 4, line 10, we added the **references** about subband [26,27], and in page 4, lines 8 ~ 10, we added the brief explanation; “Provided that **multiple subbands with step-like DOS at higher energy, which are formed by quantum confinement in two-dimensional electron gas (2DEG) systems**^{26,27}, contributed to electrical conduction,”.

Comment 3: *In view of that S-2DEG in RQW with twell of ~8 nm does not show S enhancement, is there an SDE effect in this sample?*

Answer to the comment 3: Precisely speaking, RQW with $t_{\text{well}} \sim 8$ nm has small step-like DOS effect. This step-like DOS effect is so weak that S of RQW with $t_{\text{well}} \sim 8$ nm is similar to that of 3D in scale of Fig. 3a. Misunderstanding of no step-like DOS effect comes from the description of “S-2DEG in RQW does not show S enhancement due to the large twell (Namely, equal S to the 3D GaAs)”, so we remove this sentence to avoid misunderstanding.

In page 10 line 4, we removed the sentence of “*Therein, S-2DEG in RQW does not show S enhancement due to the large twell (Namely, equal S to the 3D GaAs).*”

Comment 4: *Is it reasonable to assume that there is no contribution of ionized impurity scattering by the fact that the mobility does not vary with carrier concentration?*

Answer to the comment 4: Impurity scattering has strong carrier concentration dependence because concentration of dopant (ionized impurity) is approximately equivalent to the carrier concentration. This implies that the lack of carrier concentration dependence of carrier mobility is related to the lack of impurity scattering. So, it is reasonable, but we admit that it was too much to say that the lack of carrier concentration dependence of carrier mobility is a strong evidence of the lack of impurity scattering.

In general, the lack of impurity scattering in modulation doping structure is proved in the temperature dependence of carrier mobility. If there is no ionized impurity in 2DEG channel, the dominant scattering is polar optical phonon scattering which is weakened at lower temperature, resulting in the larger carrier mobility at the lower temperature: the orders of magnitude higher mobility at low temperature (Fig. A5a [*Appl. Phys. Lett.* **55**, 1888 (1989).]) unlike direct doped 3D bulk case with impurity scattering as shown in Fig. A5b [*Phys. Rev. Lett.* **66**, 1513 (1991).]. Therefore, the tendency of T - μ curve showing the orders of magnitude higher mobility at low temperature is a solid experimental evidence of modulation doping (the lack of impurity scattering) in this research field. On the other hand, in the present case, T - μ curve of our samples is also plotted in Fig. A5a, b, indicating that our experimental result also shows the same tendency. So, this is an experimental evidence that there is almost no ionized impurity scattering. Furthermore, our data agreed with the calculations with almost no impurity scattering as shown in Fig. A5c (the open marks).

So, we remove the sentence of “*This indicates that M-2DEG has no contribution of ionized impurity scattering inside 2DEG channel because of no impurity in the channel by MDE.*” to avoid misunderstanding and we discuss ionized impurity scattering when we mention the temperature dependence of mobility.

[43] *Appl. Phys. Lett.* **55**, 1888 (1989).

[44] *Phys. Rev. Lett.* **66**, 1513 (1991).

Fig. A5 a, b, c

In page 10, line 18, we removed the sentence of “This indicates that M-2DEG has no contribution of ionized impurity scattering inside 2DEG channel because of no impurity in the channel by MDE.”

In page 11, lines 6 ~ 11, we added the sentences of “The tendency of experimental data in the M-2DEG was explained by the theoretical T - μ curve of M-2DEG (the open marks in Fig. 3c), where the dominant scattering is polar optical phonon scattering due to the almost no ionized impurity scattering unlike 3D GaAs with ionized impurities. The orders of magnitude higher mobility at low temperature is reported as the result from modulation doping effect [43,44].”

Comment 5: An elaborate description of the measurements is recommended, including the direction of the temperature gradient when measuring the Seebeck coefficient, etc.

Answer to the comment 5: Our measurement of the Seebeck coefficient of the thin films is

conducted by using ZEM-3, where temperature difference is applied along in-plane direction, and the temperature difference and the electric voltage difference between two points on the films were obtained by thermocouple probes. We elaborated the measurement details of Seebeck coefficient.

As for the reliability of the ZEM-3 measurement, many researchers have measured the Seebeck coefficient of the thin films by using ZEM-3 and show reasonable values. As an example, we showed the case of Si, Ge and SiGe films on Si substrates as shown in Fig. A6a, b, c and we show some preceding famous studies references about thermoelectric films [*Scr. Mater.* **69**, 549 (2013)., *Adv. Mater.* **27**, 2246 (2015)., *Appl. Phys. Express* **11**, 111301 (2018)., *J. Mater. Chem. A* **7**, 17981 (2019).]. We added the references of the papers in which ZEM-3 was used for measuring S of thin films. [*Scr. Mater.* **69**, 549 (2013)., *Appl. Phys. Express* **11**, 111301 (2018).].

[45] *Scr. Mater.* **69**, 549 (2013).

[46] *Appl. Phys. Express* **11**, 111301 (2018).

In page 16, lines 3 ~ 6, we added the description of “ S was measured using ZEM-3 (ADVANCE RIKO Inc.) [45, 46], where temperature difference is applied along in-plane direction, and the differences of temperatures and the electric voltages between two points on the films were obtained by thermocouple probes.”

Fig. A6 a, b, c

Comment 6: The measured properties taken as examples in S3 are divided by unit area. How do you get the carrier concentration, S , and carrier conductivity per unit volume?

Answer to the comment 6: Carrier concentration and electrical conductivity are obtained by dividing measured sheet carrier concentration and sheet electrical conductivity by 2DEG thickness (t_{well} or t_{ch}). This definition is the same as famous previous study [*Nat. Mater.* **6**, 129 (2007)]. This is written in the section of method. Measurement details of S are written in **Answer to the comment 5**.

[20] *Nat. Mater.* **6**, 129 (2007).

In page 15, line 16 ~ page 16, line 1, we added the sentence of “**Sheet electrical conductivity and sheet carrier concentration** were measured using van der Pauw method and Hall effect measurement, respectively. **σ and n are obtained by dividing measured sheet electrical conductivity and sheet carrier concentration by t_{well} or t_{ch} [20].**”

Comment 7: The mechanism behind the Seebeck boost by M2DEG is not elucidated in this work.

Answer to the comment 7: Higher energy subbands are also occupied above the Fermi level. As a result, higher energy carriers can participate in the conduction, which causes S enhancement due to the increase of the average energy of conducting carriers like well-known energy filtering effect.

In page 4, lines 8 ~ 12, we added the above thing; “Provided that multiple subbands with step-like DOS at higher energy, which are formed by quantum confinement in two-dimensional electron gas (2DEG) systems^{26,27}, contributed to electrical conduction, S would be substantially enhanced because **the participation rate of higher-energy carriers in the carrier conduction becomes larger** (Fig. 1b and 1c);”.

Comment 8: One technical problem is that the authors used many abbreviations in the main text, which unfortunately reduces the readability of this manuscript.

Answer to the comment 8: We agreed that there are many abbreviations. To reduce the abbreviations, we do not use the abbreviations of SDE and MDE meaning step-like DOS effect and modulation doping effect.

In the manuscript, we revised “SDE” to “**step-like DOS effect**”, and “MDE” to “**modulation doping effect**.”

Comment 9: In addition, there is a lack of definitions of the used terms and abbreviations, for example, TQW and RQW refer to triangular quantum well and rectangular quantum well, respectively. An explanation of which quantum well is triangular or rectangular in the beginning they appear will enhance the readability of this manuscript.

Answer to the comment 9: TQW and RQW are defined when they appeared at first.

Abbreviations E_i and E are also defined when they appeared at first. Furthermore, we admit that abbreviation E_c should be defined in main text.

In page 5, lines 1 ~ 3, TQW and RQW were defined.

In page 6, lines 2 ~ 3, we defined E_c as conduction band bottom of 3D GaAs, E as carrier energy, and E_i as the bottom energy of i -th subband.

Comment 10: *“Some studies achieved 100-200 times reduction of kappa, making a big impact on TE research”. This sentence might incur a misunderstanding.*

Answer to the comment 10: Thank you for your pointing out the sentence. We changed from “100-200 times reduction of kappa” to “100-200 times smaller κ by introducing nanostructures.”

In page 3, line 8, we revised the sentence from “100-200 times reduction of kappa” to “**100-200 times smaller κ by introducing nanostructures.**”

In addition, we revised some words to improve English grammar and to increase readability for readers. Revised words are shown with green highlight. Furthermore, description of the citation of supplementary information, subheadings, and abstract were revised based on the formatting instructions, we revised some words, which were marked by blue light highlight. The revisions written in the above-mentioned “answer to the comments” were marked by yellow highlighted.

Reviewer #3:

Comment: *The manuscript presents an interesting approach with promising results, but several key aspects need to be addressed and clarified before considering acceptance. Specifically, the authors should correct writing issues, provide a review of M2DE, clarify the formation of M-2DEG and S-2DEG, provide additional experimental evidence, and include missing details on Hall measurement, ionized impurity scattering, and simulation assumptions. Detailed comments are as follows:*

Answer to the comment: Thank you for your comment. We are trying to address and clarify the issues the reviewer pointed out as faithful as possible by answering the following reviewer's comments.

Comment 1: *Writing Issues: There are several writing issues that need to be corrected:*

Comment 1-a: *In equations 1 and 2, "where E_i is the subband energy at the i th band" is mentioned, but there is no E_i in either equation. Do the authors mean that 'E' in equation 1 is the subband energy at the i th band?*

Answer to the comment 1-a: 'E' is carrier energy, and ' E_i (E_i)' in Eq. 1 is the bottom energy of i -th subband. In Eq. 1, " E_i " was used in the integration range in the previous manuscript, but the description was wrongly " E_i ". The " E_i " should be revised to " E_i ". In the revised manuscript, we modified from ' E_i ' to ' E_i '. We moved the places of definitions of 'E' and ' E_i ' to the same place to distinctify these definitions. Furthermore, R_0 (=Eq. 1/Eq. 2) is a function of this ' E_i ', so, it is mentioned in the revised manuscript.

In Eq. 1, in page 7, line 11, we revised " E_i " to " E_i ."

From page 7, line 13 to page 6, line 3, we moved the definitions of " E_i ," and from page 7, line 4 to page 6, line 2, we moved the definition of "E."

In page 7, lines 14 ~ 15, we added the sentence of "**It is found that R_0 is the function of E_i from Eq. 1 and 2.**"

Comment 1-b: *Some descriptions in the manuscript are not scientific, for example, "In RQW, the energy difference between discrete subbands at higher energy is quite larger than that between*

*the lowest level in RQW and conduction band bottom of 3D GaAs." The term "quite larger" is vague. Authors should provide a quantitative description, such as "*** eV higher than ***".*

Answer to the comment 1-b: We agreed that vague description is not appropriate. In this description, we would like to say that the energy differences between the discrete subband bottoms at higher energy in RQW is increasing with increase of i value. So, one or two subbands can only exist in RQW. We revised the sentence to avoid ambiguous description.

In page 6, lines 4 ~ 9, we revised the sentences of “**In general RQW, the energy difference between discrete subband bottoms ($E_{i+1}-E_i$) is monotonically increasing with increase in the i value. Therefore, unlike TQW, it is expected that one subband (or two subbands) can only exist in the present AlGaAs/GaAs/AlGaAs RQW with ~ 0.2 eV barrier height when the step-like DOS appears due to the sufficiently small t_{well} , indicating S-2DEG system (Supplementary Note 2).**”

Comment 2: *Lack of Review on M2DE: The authors claim the presence of the "multiplied two-dimensional electron gas effect (M2DE)". This appears to be based on quantum confinement and the formation of subbands in two-dimensional electron gas (2DEG) systems, which are well-studied phenomena. However, as a major novelty claimed, there is no review about M2DE in the introduction. A brief description of M2DE with references is necessary.*

Answer to the comment 2: M2DE means “*S enhancement effect*” via multiple subbands in 2DEG, and this is new strategy we proposed. On the other hand, as the reviewer mentioned, multiple subbands in two-dimensional electron gas (2DEG) systems itself is well-known. So, as the reviewer mentioned, we revised the sentence by inserting the brief description of multiple subbands in two-dimensional electron gas (2DEG) systems with two famous references [26, 27] as follows.

[26] *Rev. Mod. Phys.* **54**, 437 (1982).

[27] *Phys. Rev. B* **53**, R10493 (1996).

In page 4, lines 8 ~ 12, we added the above thing; “**Provided that multiple subbands with step-like DOS at higher energy, which are formed by quantum confinement in two-dimensional electron gas (2DEG) systems [26, 27],** contributed to electrical conduction, S would be substantially enhanced because the participation rate of higher-energy carriers in the carrier

conduction becomes larger (Fig. 1b and 1c);”.

Comment 3: Clarification on M-2DEG and S-2DEG Formation: The authors claim the formation of M-2DEG and S-2DEG in TQW and RQW samples, respectively. It is suggested to highlight the structure difference of these two samples by providing a figure.

Answer to the comment 3: We agree that the structures and the band diagrams of RQW and TQW will be helpful for understanding the difference between TQW and RQW in main text. We added the simple version of stacked layered structures of RQW and TQW with their band diagrams (Fig. A7a, b) to Fig. 2 in main text.

Fig. A7

In Fig. 2, simple version of stacked layered structures of RQW and TQW with their band diagrams was shown.

In page 6, lines 1 ~ 4, we added the sentence of “**Illustrations of sample structures and simple band diagrams are shown in Fig. 2a, and 2b, where conduction band bottom of 3D GaAs (E_c), carrier energy (E) and the bottom energy of i -th subband (E_i). The index i ($i=1, 2, \dots$) is the subband number, where the subband bottom with the smaller number of i positions at the lower energy level.**”

Comment 4: *Lack of Experimental Evidence: In figure 2a, the authors claim the distribution of multiple subbands with a gap around 0.02 eV. This claim, if true, may lead to the enhancement of the Seebeck coefficient at 300K (corresponding to $k_B T$ of 25.9 meV). However, apart from the measured Seebeck at 300K, there is a lack of other experimental evidence to support the claim. A temperature dependence measurement of the Seebeck coefficient would be more convincing to prove the presence of M-2DEG. This measurement offers modulation of the responding energy range of around $2 k_B T$, which is sensitive enough for multiple subbands with a gap around 0.02 eV.*

Answer to the comment 4: As the reviewer mentioned, the temperature dependence of Seebeck coefficient is important to prove our proposed mechanism. Newly, we carried out both the calculation and experiments about the temperature dependence of Seebeck coefficient and occupation ratio of M-2DEG, as shown in Fig. A8a-c. Figure A8a shows the calculated T - S curve, which shows a gradual change even around the temperature corresponding to the energy difference $E_2 - E_1 (=k_B T)$. It might be expected that temperature dependence of Seebeck coefficient shows a drastic change at the temperature corresponding to the energy difference $E_2 - E_1 (=k_B T)$, but according to the calculation, it is actually changed gradually. The reason is mentioned as follows; the subband bottom is discrete, but the subband itself has a continuous energy due to the energy at the x and y directions. Therein, carriers distribute in this step-like but continuous density of states in accordance with the Fermi-Dirac function. Step-like DOS is sharp at E_2 , but the change of the product of DOS and the Fermi-Dirac distribution tail is small (Fig. A8d). As a result, change of Seebeck coefficient coming from step-like DOS change at E_2 is too small to detect (Fig. A8a). The carrier occupation ratio R_O was defined as $R_O = n_i / n_i$, where n_i is sheet carrier concentration at the i -th subband with the bottom energy E_i and n_i is the sum of n_i . Therein, R_O of the first subband at 160 K is larger than that of 300 K (Fig. A8b). However, some carriers exist in higher subbands even at low temperature because n_i gradually reduces based on the temperature dependence of Fermi-Dirac distribution. Therefore, M2DE is gradually suppressed at low temperature. Figure A8c shows the experimental results of temperature dependence of the Seebeck coefficient along

with the calculated results. This demonstrates that the calculation and experimental results agreed well. This proved the existence of M2DE in terms of temperature dependence.

In this way, we successfully obtained the agreement between experimental T - S curve and theoretical one with M2DE by measuring T - S thanks to the reviewer's comment. It is interesting that M2DE is proved by the temperature dependence. We added the above discussion to the supplementary information. Thank you.

In page 10, lines 9 ~ 10, we added the words of "Furthermore, the S calculation (the open yellow triangles) including M2DE in M-2DEG agreed with the experimental n - S data (Fig. 3a) and T - S data (Supplementary Note 6),".

In supplementary information, we added the section (supplementary information 6) of "Temperature dependence of the Seebeck coefficient of TQW."

Fig. A8 a, b, c d

Comment 5: *Missing Details on Hall Measurement: Details about the Hall measurement are missing. The authors claim the measurement of samples with relatively low carrier concentration of 10^{17} to 10^{18} cm^{-3} , which is challenging. Information about the capability of the instrument (e.g., magnetic field, SMU information) is necessary. Also, the error should be stated somewhere, for example, in the legend of figure 3.*

Answer to the comment 5: In our Hall effect measurement, the errors of n and μ are about 13%. We used 2401 sourcemeter (Keithley) as SMU, and the range of magnetic field is from -0.5 T to 0.5 T. In the case of semiconductor, it is easy to measure under this condition. We added the information in the revised manuscript.

In page 16, lines 1 ~ 3, we added the sentences of “**In our Hall effect measurement, we used 2401 sourcemeter (Keithley) as source measure unit, and the range of magnetic field is from -0.5 T to 0.5 T. Therein, the errors of n and μ are about 13%.**”

Comment 6: *Lack of Evidence on Absence of Ionized Impurity Scattering: The authors state that "Interestingly, μ values of M-2DEG hardly depended on the n , while the calculation μ curve of 3D GaAs exhibited a monotonic decrease as n increased. This indicates that M-2DEG has no contribution of ionized impurity scattering inside the 2DEG channel because of no impurity in the channel by MDE." The authors must provide solid experimental evidence to prove the absence of ionized impurity scattering. If the M-2DEG effect is present, the measured mobility is an average mobility of multiple bands, which may have varied effective masses, thus causing the observed effect in mobility. Have the authors considered this possibility, and how did they exclude it?*

Answer to the comment 6: Impurity scattering has strong carrier concentration dependence because concentration of dopant (ionized impurity) is approximately equivalent to the carrier concentration. This implies that the lack of carrier concentration dependence of carrier mobility is related to the lack of impurity scattering. But, we admit that it was too much to say that the lack of carrier concentration dependence of carrier mobility is a strong evidence of the lack of impurity scattering because modulation doping is judged from the temperature dependence of carrier mobility in general.

In general, the lack of impurity scattering in modulation doping structure is proved in the

temperature dependence of carrier mobility. If there is no ionized impurity in 2DEG channel, the dominant scattering is polar optical phonon scattering which is weakened at lower temperature, resulting in the larger carrier mobility at the lower temperature: the orders of magnitude higher mobility at low temperature (Fig. A9a [*Appl. Phys. Lett.* **55**, 1888 (1989).]) unlike direct doped 3D bulk case with impurity scattering as shown in Fig. A9b [*Phys. Rev. Lett.* **66**, 1513 (1991).]. Therefore, it is considered that this tendency of T - μ curve showing the orders of magnitude higher mobility at low temperature is a solid experimental evidence of modulation doping (the lack of impurity scattering). This time, in the present case, we newly analyzed T - μ data of our samples in more detail, resulting in the solid conclusion as follows. Our data are also plotted in Fig. A9a, b, indicating that our experimental results of TQW and 3D GaAs also show the same tendency as those of modulation doped GaAs without ionized impurity scattering and direct doped 3D GaAs bulk with impurity scattering, respectively. So, this is an experimental evidence that there is almost no ionized impurity scattering. Furthermore, our data agreed with the calculations with almost no impurity scattering as shown in Fig. A9c (the open marks).

In addition, the change in effective mass could be caused by non-parabolicity of GaAs, which is reported to be very small of ~4% even at 29.1 meV higher than subband bottom [*Phys. Rev. B* **43**, 11787 (1991).]. Actually, in Hirakawa's famous study of carrier mobility in modulation doped 2DEG-GaAs, constant effective mass without non-parabolicity were used [*Phys. Rev. B* **33**, 8291 (1986).]. Therefore, the effective mass change effect can be ignored compared with the measurement error. To prove negligibility of this effect, we newly performed calculation experiments, as shown in Fig. A10, where we calculated the mobility in our TQW by considering the effective mass change from non-parabolicity. The calculation results show that the mobility of M-2DEG (green triangles) is changed by ~6% from calculated mobility using parabolic band with constant effective mass (yellow triangles). This demonstrated that this change is small enough compared with the measurement error, and that it does not affect the tendency of n - μ curve. So, the effective mass change effect cannot be a cause of independence of μ on the n .

In summary, historically, it is considered that the tendency of T - μ curve is a solid experimental evidence for modulation doping. Therefore, we removed the sentence about the lack of carrier concentration dependence meaning the lack of impurity scattering. Instead, we discuss about the lack of ionized impurity scattering at the place where the temperature dependence of mobility is shown in the manuscript. In addition, we added the references about the small change in effective mass of GaAs to "Numerical calculation".

[43] *Appl. Phys. Lett.* **55**, 1888 (1989).

[44] *Phys. Rev. Lett.* **66**, 1513 (1991).

[51] *Phys. Rev. B* **43**, 11787 (1991).

In page 10, line 18, we removed the sentence of “This indicates that M-2DEG has no contribution of ionized impurity scattering inside 2DEG channel because of no impurity in the channel by MDE.”

In page 11, lines 6 ~ 11, we added the sentences of “The tendency of experimental data in the M-2DEG was explained by the theoretical T - μ curve of M-2DEG (the open marks in Fig. 3c), where the dominant scattering is polar optical phonon scattering due to the almost no ionized impurity scattering unlike 3D GaAs with ionized impurities. The orders of magnitude higher mobility at low temperature is reported as the result from modulation doping effect [43,44].”

In page 19, lines 1 ~ 3, we added the sentence of “In the calculation, m value shown in Table 1 was used for each subband under the assumption that non-parabolicity effect on m is negligible [51].”

Fig. A9 a, b, c

Fig. A10

Comment 7: *Missing Simulation Details:* The authors state that "The tendency of experimental data in the M-2DEG was well explained by the theoretical T - μ curve." However, the details of the simulation are missing. For example, what are the assumptions for the scattering mechanism of the plotted theoretical T - μ curve?

Answer to the comment 7: In the section of numerical calculation, we show polar optical phonon scattering, acoustic deformation potential scattering, remote ionized impurity scattering, interfacial roughness scattering is considered in calculation. This information is written in section of "Numerical calculation" and used parameters are written in table 1. But this information might be difficult for readers to find at the place of discussion about theoretical T - μ curve. Therefore, at the place at the beginning of TE properties in main text, we added the above information in the revised manuscript.

In page 9, lines 2 ~ 6, we revised the sentences of "Experimental **and calculated** TE properties of M-2DEG and S-2DEG are shown in Fig. 3. Therein, theoretical calculation of S and μ was performed under parabolic band and relaxation time approximations on the basis of Boltzmann transport theory⁴¹. **The details of carrier scattering models and used parameters are written in the section of "Numerical calculation" and table 1 respectively.**"

In addition, we revised some words to improve English grammar and to increase readability for readers. Revised words are shown with green highlight. Furthermore, description of the citation

of supplementary information, subheadings, and abstract were revised based on the formatting instructions, we revised some words, which were marked by blue light highlight. The revisions written in the above-mentioned “answer to the comments” were marked by yellow highlighted.

REVIEWER COMMENTS

Reviewer #1 (Remarks to the Author):

The authors carefully and thoughtfully revised the manuscript following the reviewers comments. I recommend publishing the revised manuscript.

Reviewer #2 (Remarks to the Author):

Thanks for the response. The answer to these questions gives a better understanding of this work. However, there are still two issues that prohibit me from giving a positive recommendation.

1. Even though the author has responded to comment 2, the meanings of the 'E_C' line and the several violet lines in Figure 2a are still confusing. What's the basis of the 'E_C' line? Since the energy of the subbands is independent of z, why does the energy of E_C increase with z increasing? Besides, as the author says, "Subband' is the energy band in two-dimensional carrier gas system, where the bottom energy of each band is discrete energy level caused by quantum confinement at the z direction". Why no subbands in RQW, since the width of the RQW is similar to TQW? If there were subbands in RQW, what does multiple-2DEG indeed refer to? To answer these questions, I guess the author can make a better clarification on M-2DEG and S-2DEG formation, which the other two reviewers care about, too.

2. I don't agree with the author about the discussion of the scattering mechanisms in response 4. For degenerate semiconductors, the mobility decreases with carrier concentration increasing when acoustic phonon scattering (APS) or alloy scattering (AS) dominates. It's unreasonable to conclude no contribution of ionized impurity scattering from the fact that the mobility is independent of the carrier concentration. As the tendency of $T-\mu$ is better evidence to prove no impurity scattering, I would like to recommend to remove of the discussion of the carrier concentration dependence of mobility.

Reviewer #3 (Remarks to the Author):

Thanks to the comprehensive elucidation and the inclusion of supplemental data in both the main text and SI. The enhanced clarity regarding experimental methodologies, data interpretation, and the corroboration of their assertions significantly aids in grasping the outcomes of this study. The augmented explanations not only enhance precision but also bolster their claims with the supplementary data. Given that the revised manuscript is more refined and lucid, I would advocate for its acceptance in Nature Communications.

Reviewer comments and Point-by-point response

Reviewer #1:

Comment: *The authors carefully and thoughtfully revised the manuscript following the reviewers comments. I recommend publishing the revised manuscript.*

Answer to the comment: Thank you for your comment.

Reviewer #2:

Comment: *Thanks for the response. The answer to these questions gives a better understanding of this work. However, there are still two issues that prohibit me from giving a positive recommendation.*

Answer to the comment: Thank you for your comment. We are trying to solve the two issues the reviewer pointed out as faithful as possible by answering the following reviewer's comments.

Comment 1-1: *Even though the author has responded to comment 2, the meanings of the 'E_C' line and the several violet lines in Figure 2a are still confusing. What's the basis of the 'E_C' line? Since the energy of the subbands is independent of z, why does the energy of E_C increase with z increasing?*

Comment 1-2: *Besides, as the author says, "Subband' is the energy band in two-dimensional carrier gas system, where the bottom energy of each band is discrete energy level caused by quantum confinement at the z direction". Why no subbands in RQW, since the width of the RQW is similar to TQW? If there were subbands in RQW, what does multiple-2DEG indeed refer to? To answer these questions, I guess the author can make a better clarification on M-2DEG and S-2DEG formation, which the other two reviewers care about, too.*

Answer to the comment 1-1: We explain about E_C ($E_{_C}$). E_C , conduction band bottom, is changed by the electrical charge based on the Poisson's equation. This is the reason of "*the energy of E_C increases with z increasing.*" The way of thinking about energy band diagram (E_C line) is the same both in TQW and in RQW. We are explaining it by using the following figure (Fig. A1a) that is the illustration of energy band diagram of RQW. In undoped substrate and buffer layer, which are on the right side in Fig. A1a, E_C line is flat (*the basis of the 'E_C' line*). In barrier AlGaAs near the channel GaAs and on the buffer side, E_C is a little bended by negative charge in 2DEG. E_C in GaAs is lower than that in AlGaAs due to the band offset ΔE_C between AlGaAs and GaAs ($\Delta E_C = E_{C_AlGaAs} - E_{C_GaAs}$). Therefore, in channel GaAs, carriers are confined due to this band offset barrier to form 2DEG. In AlGaAs spacer, the slope of the E_C is constant because there is no charge in AlGaAs spacer. In doped AlGaAs, E_C is bended by the positive charge of donor ions. In undoped cap GaAs, the slope of the E_C is constant and the surface state causes high E_C at surface. Such an explanation is also applicable for TQW in Fig. 2b in main text. This explanation about the band diagram of quantum well is written in several textbooks about semiconductor: for examples, (1) "Claude Weisbuch and Borge Vinter, Quantum Semiconductor Structures:

Fundamentals and Applications (Academic Press, San Diego, 1991).”, (2) “Davies, J. H. The Physics of Low-Dimensional Semiconductors: An Introduction (Cambridge University Press, Cambridge, England, 1997).”, and so on. According to the reviewer’s comments, for readers to understand the band diagrams easier, we newly revised the manuscript, and the detail of its revision is written below (**Revisions corresponding to the answers to the comments 1-1 and 1-2.**).

Answer to the comment 1-2: We explain about subbands. At first, let’s consider infinite RQW as a simple model. Therein, the bottom energy of i -th subband (E_i) is described as follows: $E_i = \hbar^2(i\pi/t_{\text{well}})^2/(2m)$. It represents that $(E_{i+1}-E_i)$ becomes larger with increasing “ i ” in RQW because second order differential of E_i with i is positive, so E_i becomes much larger with slightly increasing “ i ”. Next, let’s consider the finite well for our RQW samples dealt with in this study. Therein, energy barrier height is finite, and equal to band offset ΔE_C . When number i is slightly increased, E_i easily becomes larger than energy barrier height. Therefore, i -th subband with slightly increased i cannot exist in the well. So, only one or two subbands with smaller E_i ($i=1$ or 2) than the energy barrier height of finite well (band offset ΔE_C) exist as shown in Fig. A1b. In the case of our RQW with $t_{\text{well}}=3-6$ nm, there is only one subband ($i=1$). On the other hand, in the case of our RQW with $t_{\text{well}}=8-12$ nm, there are two subbands ($i=1$ and 2). But, even when two subbands exist, there is a large energy difference between first subband and second one (E_2-E_1). As a result, the carrier occupation ratio of second subband is negligible in finite RQW, as shown by purple marks in Fig. A1c. (Namely, contribution of second subband is negligible.) Therefore, “single-2DEG” refers to 2DEG system with one or two (almost one) subbands as shown by purple marks in Fig. A1c. On the other hand, “multiple-2DEG” refers to the 2DEG system where many subbands exist and energy distance between subbands are small as shown by red marks in Fig. A1c, which is described as below.

In the case of TQW, E_i in ideal triangular well potential is described as follows: $E_i = (3\pi(i-1/4)/2)^{2/3} \times ((eF\hbar)^2/(2m))^{(1/3)}$, as shown in Fig. A1b. In contrast to the case of RQW, it represents that $(E_{i+1}-E_i)$ becomes smaller with increasing “ i ” in TQW because second order differential of E_i with i is negative. It should be noted that this is main origin of the difference of the number of subbands that can exist in the well of TQW and RQW. Namely, E_i with large i in TQW becomes much smaller than that of RQW at the same i . Then, in TQW, this leads to larger number of subbands with smaller E_i than energy barrier of TQW as shown in Fig. A1b.

For the reason we explained above, at similar confinement width (8-12 nm), 1D Poisson calculation demonstrates the number of the subbands that can exist in the well is 2 for RQW (small number) and ≥ 17 for TQW (large number).

As the reviewer mentioned, the clarification on M-2DEG and S-2DEG formation can support the

distinction between S-2DEG in RQW and M-2DEG in TQW. So, we added the supplementary note saying the above things.

Fig. A1

Revisions corresponding to the answers to the comments 1-1 and 1-2.

According to the reviewer's comments, for readers to understand the band diagrams easier, we newly added the explanations to supplementary note 2 in supplementary information and the reference about energy band diagram of the quantum well in main text as follows.

At the beginning of supplementary note 2 in supplementary information, we added the explanation saying E_C change in energy band diagram is based on the Poisson's equation, and in the caption of Fig. 2 in main text, we also added the reference of one of the textbooks about energy band diagram of the quantum well along with the illustration of simplified energy band diagram in Fig. 2a.

In supplementary note 2 in supplementary information, we also added the saying the explanation about the number difference of the subbands that can exist in RQW and TQW and the clarification about S-2DEG and M-2DEG. Based on this addition, we changed the name of Supplementary Note 2 to “Energy band diagrams of rectangular quantum well: single two-dimensional electron gas system and the clarification of single and multiple two-dimensional electron gas systems.”

The revisions in Supplementary Information are marked by yellow highlights in “GaAs_highlighted_SI_2nd.pdf”.

Comment 2: *I don't agree with the author about the discussion of the scattering mechanisms in response 4. For degenerate semiconductors, the mobility decreases with carrier concentration increasing when acoustic phonon scattering (APS) or alloy scattering (AS) dominates. It's unreasonable to conclude no contribution of ionized impurity scattering from the fact that the mobility is independent of the carrier concentration. As the tendency of $T-\mu$ is better evidence to prove no impurity scattering, I would like to recommend to remove of the discussion of the carrier concentration dependence of mobility.*

Answer to the comment 2: As the reviewer mentioned, $T-\mu$ is better evidence, while carrier concentration dependence of carrier mobility is not direct evidence. So, we remove the sentences in $n-\mu$ discussion part as follows.

In page 10, line 16 in revised manuscript (in page 10, lines 16 ~ 18 in “GaAs_highlighted_NC_2nd.pdf”), we removed the sentence of “Interestingly, μ values of M-2DEG hardly depended on the n , while the calculation μ curve of 3D GaAs exhibited monotonic decrease as n increased.”

In page 10, line 18 in revised manuscript (in page 11, lines 1 ~ 3 in “GaAs_highlighted_NC_2nd.pdf”), we removed the sentence of “This agreement supports that modulation doping effect, step-like DOS effect, and M2DE appear in M-2DEG in TQW.”, and revised the words of “confirmed this from” to “obtained.”

In addition, we revised some words to improve English grammar and to increase readability for readers. Revised words are shown with green highlight in “GaAs_highlighted_NC_2nd.pdf”. The revisions written in the above-mentioned “answer to the comments” were marked by yellow highlighted in “GaAs_highlighted_NC_2nd.pdf” for main text and in “GaAs_highlighted_SI_2nd.pdf” for Supplementary Information.

Reviewer #3:

Comment: *Thanks to the comprehensive elucidation and the inclusion of supplemental data in both the main text and SI. The enhanced clarity regarding experimental methodologies, data interpretation, and the corroboration of their assertions significantly aids in grasping the outcomes of this study. The augmented explanations not only enhance precision but also bolster their claims with the supplementary data. Given that the revised manuscript is more refined and lucid, I would advocate for its acceptance in Nature Communications.*

Answer to the comment: Thank you for your appreciation of our manuscript.

REVIEWERS' COMMENTS

Reviewer #2 (Remarks to the Author):

The revision can be published as is.

Reviewer comments and Point-by-point response

Reviewer #2:

Comment: *The revision can be published as is.*

Answer to the comment: Thank you for your appreciation of our manuscript.